# Unveiling the Hidden Rules of Spherical Viruses Using Point Arrays

**DOI:** 10.3390/v12040467

**Published:** 2020-04-20

**Authors:** David P. Wilson

**Affiliations:** Department of Physics, Kalamazoo College, Kalamazoo, MI 49006, USA; David.Wilson@kzoo.edu; Tel.: +1-(269)-337-7096

**Keywords:** protruding features, spherical virus, point arrays, surface modifications, virus-like particle, VLP, drug delivery, icosahedral, nanomedicine, ligand binding

## Abstract

Since its introduction, the Triangulation number has been the most successful and ubiquitous scheme for classifying spherical viruses. However, despite its many successes, it fails to describe the relative angular orientations of proteins, as well as their radial mass distribution within the capsid. It also fails to provide any critical insight into sites of stability, modifications or possible mutations. We show how classifying spherical viruses using icosahedral point arrays, introduced by Keef and Twarock, unveils new geometric rules and constraints for understanding virus stability and key locations for exterior and interior modifications. We present a modified fitness measure which classifies viruses in an unambiguous and rigorous manner, irrespective of local surface chemistry, steric hinderance, solvent accessibility or Triangulation number. We then use these point arrays to explain the immutable surface loops of bacteriophage MS2, the relative reactivity of surface lysine residues in CPMV and the non-quasi-equivalent flexibility of the HBV dimers. We then explain how point arrays can be used as a predictive tool for site-directed modifications of capsids. This success builds on our previous work showing that viruses place their protruding features along the great circles of the asymmetric unit, demonstrating that viruses indeed adhere to these geometric constraints.

## 1. Introduction

We present a modified fitting method for classifying spherical viruses using the icosahedral point arrays introduced by Keef and Twarock [1,2]. These point arrays reveal an interdependence between the location of viral protrusions, capsid proteins and the geometric packaging of the viral genome. We offer new interpretations of these point arrays as geometric restrictions that place constraints on which amino acids are possible to modify. We also discuss how these point arrays might be used to determine new potential locations for significant external or internal capsid modifications such as decoration proteins or ligand binding.

Spherical viruses have been canonically well described by their Triangulation number (*T*-number), introduced by Caspar and Klug [3], which posits that capsid proteins are arranged in such a manner as to have nearly identical chemical environments, known as quasi-equivalence. In general, the *T*-number specifies the total number of proteins needed (60T) to form the capsid, as well as the *T* proteins within the asymmetric unit (Figure 1) and that there are 12 pentamers and 10(T−1) hexamers subunits in a capsid. While there are several notable exceptions to these rules, each of these exceptions is arranged in the same subunit layout as the canonical *T*-numbers. For example L-A virus, is a 120 protein capsid (T=2) with the same architecture as a T=1 capsid, though it has 10 proteins making up its pentameric unit and no hexamers. Also of note is SV40, which has a T=7d architecture. SV40 is composed entirely of pentameric subunits, with 60 pentamers residing where the 60 hexamers would normally be located, reducing the capsid from 420 proteins to 360 proteins. The *T*-number is also able to describe capsids with pseudo-Triangulation (pT) numbers, which are composed of multi-domain proteins which form pseudo-pentamers and pseudo-hexamers to mimic larger *T*-number capsids. CPMV is an example of a pT3, which has the same subunit architecture as a T=3 capsid, appearing to be composed of 12 pentamers and 20 hexamers, though only containing 60 proteins instead of the expected 180 proteins. While the *T*-number is a powerful description, it is fundamentally a limited tool as it uses a 2d thin-shell description that only specifies the general protein arrangement of the viral capsid, and does not specify where or how capsid proteins are oriented, nor their radial distribution nor any information about the organization of the genetic material contained within.

It has been demonstrated that several spherical viruses conform to the geometric constraints of the icosahedral point arrays introduced by Keef and Twarock et al. [1,2]. The points of these arrays represent material boundaries at multiple radial levels for proteins, genetic material as well as their interfaces. Each point array imposes a different set of radial and angular constraints on the virus [1,2,4]. The geometry of viral architecture has also been explored by Janner [5,6,7,8,9] using encasing forms which are constructed for viral components at different radial levels by embedding virus structure into lattices. Janner has also explored fitting point arrays by hand to viral capsids, and has argued that a more specific toolkit should be developed for the analysis of the viral architecture, including comparisons with indexed backbone positions and regions with site specific point symmetry [10]. We showed that viruses place their protruding features at discrete gauge points along the 15 icosahedral great circles which are used to create the asymmetric unit (Figure 1), a direct consequence of viruses conforming to the geometric constraints of point arrays [4]. These gauge points indicate to the overall scaling of the point array. Another promising method of generating icosahedral point arrays has been offered by Zappa et al. [11,12]. They propose generating point arrays from 6d projections of the hypercube. Our point array fitting methods presented in this paper are compatible with this scheme, if a systematic library can be developed. Point arrays are a natural tool for measuring and characterizing spherical capsids, as they can be used to track changes throughout maturation as well as to quantify individual features which differ between capsids. In recent years, additional capsid descriptions have been offered, including an expanded Triangulation number by Rochal et al. [13] and icosahedral surface tiling by Twarock et al. [14], though each of these schemes focuses on surface organization rather than a detailed radial description.

In this work we will present an improved fitness algorithm which uses gauge point agreement of protruding features to determine the potential fits to the 55 single and 514 double point arrays developed by Keef and Twarock [1,2], to characterize the viral architecture of spherical viruses. This improved algorithm was needed to clarify and remove poorly matching arrays that were found to fit viruses in Refs. [2,15]. Through gauge point filtering and removal of several free parameters, we have greatly clarified the fits of these point arrays. We will also show how to use these point array fits to suggest surface sites where virus capsids should and should not be modified, independent of their local chemistry, including the relative reactivity of surface lysine residues of the (pT3) CPMV [16,17]. We will demonstrate that the location of protruding features dramatically limits the possible interior structure of viruses, including where to place interior drug compounds for potential therapeutic virus-like particles (VLPs).

## 2. Materials and Methods

We begin by reviewing icosahedral symmetry and the construction of the 55 icosahedral point arrays generated by affine extensions of the base icosahedral polyhedra developed in the seminal papers by Keef and Twarock [1,2]. A point array is essentially a collection of points at one or more radii with precise angular locations on a sphere and a fixed ratio between their radii. The motivation for building these point arrays is to create a mathematically consistent representation of icosahedral symmetry at multiple radial levels. While icosahedral symmetry is the largest discrete rotation symmetry of a sphere, it makes no distinction based on radii. Even T=1 viruses, composed of only 60 proteins arranged into 12 pentamers, can have dramatically different expressions of this symmetry at multiple radial levels (Figure 2). In contrast to the *T*-number, point arrays provide relative radial locations for capsid proteins and genetic material which are specified by a single free parameter, the global radial scaling. The points of these arrays represent the locations of material boundaries and key structural locations. Encoded within these point arrays are limits on the maximum relative distance and orientation of the capsid protein features. In some cases, these limits will also impose constraints on packing of the genetic material contained within.

### 2.1. Icosahedral Symmetry

The chiral icosahedral rotation symmetry group (I) is a set of 60 rotations consisting of 2-fold, 3-fold and 5-fold rotations on a sphere, and is the largest discrete finite point group without reflections (modulo spatial inversions). Reflections are not considered due to the chirality of the amino acids which make up proteins. Spherical capsids may be subdivided into 60 equivalent sections, known as the asymmetric unit (AU), similar to a unit cell in crystallography (Figure 1). Due to the reducibility of the full capsid to the AU, we will often express our results in terms of it. The group of 60 icosahedral rotation matrices can be generated by successive combinations of the 2-fold rotation a and the 3-fold rotation b which border the asymmetric unit (Figure 1), as
(1)$$I\equiv \langle a,b|a^{2}=b^{3}={\left(ab\right)}^{5}=1\rangle .$$

These rotation matrices and all our polyhedral vertices are given in the Appendix A.

We begin with the vertices of the three standard (base) representations of icosahedral symmetry the icosahedron (ICO), Dodecahedron (DOD) and Icosadodecahedon (IDD) which are representations of the 5, 3 and 2-fold symmetries, respectively (Figure 3). The vertices of these polyhedra will also serve as our translation vectors for the affine extensions below. We align all of our structures with the Viper Database orientation [21] of the spherical volume with a 2-fold axis aligned with the +z direction and a 5-fold aligned with the vector (0,1,ϕ), where $\phi =\frac{1+\sqrt{5}}{2}\approx 1.618$ is the golden ratio. The vertices of each of the three respective polyhedra are all equidistant from the origin and therefore constitute only a single radial level.

Throughout this work we will show images of point arrays layered atop virus capsids. We have introduced a standard color scheme, inspired by the icosahedral building kits made by Zometool. The symmetry axes have primary colors 5-fold (red), 3-fold (yellow) and 2-fold (blue). The points along the great circles connecting neighboring symmetry axes (Figure 1) are based on the paint color addition of those two colors, see Table 1.

### 2.2. Affine Extensions

The construction of the 55 point arrays begins with translating a base polyhedron ICO, DOD or IDD in a 2-fold $\left(\lambda {\vec{T}}_{2}\right)$, 3-fold $\left(\lambda {\vec{T}}_{3}\right)$ or 5-fold $\left(\lambda {\vec{T}}_{5}\right)$ direction (Figure 3), scaled by an amount λ. Translations which result in the intersection of displaced vertices are deemed admissible affine extensions [1,2]. This intersection reduces the cardinality (number of elements) of the point array from the maximal free group representation. The translated set of points no longer has icosahedral symmetry at each radial level. We restore the symmetry by applying the 60 icosahedral rotations to form the complete point array (Figure 4). Each point arrays can be considered to have two components, the base vertices and the point cloud generated by applying icosahedral symmetry to the affine extension. Affine extensions are linear transformations which preserve parallel relationships within a geometric representation; however, they do not preserve local angles or distances and are a standard tool for extending group symmetry.

For example, $\lambda =\frac{1}{\phi }$ is an admissible scaling length for $\lambda {\vec{T}}_{5}$ of ICO, which results in an intersection of displaced vertices at a 3-fold axis, thus reducing the cardinality. This process is illustrated in Figure 4. We represent this point array as $\mathrm{ICO}\cup I(\mathrm{ICO}+\phi {\vec{T}}_{5})$, which is the union (∪, combination) of the base polyhedron ICO and the point cloud generated by the 60 icosahedral rotations I of the translated vertices $\mathrm{ICO}+\phi {\vec{T}}_{5}$.

This approach might seem ill-conceived, as icosahedral symmetry is the largest compact rotation symmetry in 3d. However, this method does not add any new symmetry axes, rather it creates a representation of icosahedral symmetry at multiple radial levels. These representations have specified ratios of radial levels with only one free parameter, overall scaling of the point array. It has been suggested that this extended symmetry could help explain the inherent stability of viral capsids [2].

The first 41 point arrays [1] are created when these translated vertices intersect at icosahedral symmetry axes, leading to arrays with additional polyhedra representations of 2-, 3- or 5-fold symmetry, now at two or more radial levels (Figure 5). The additional 14 point arrays [2] are created from intersections near, though not on symmetry axes. This means that they do not create any new icosahedral polyhedra at different radial levels, though they still reduce the cardinality from the free group representation. These point arrays are representations of the group symmetry across multiple radial levels. It can readily be shown that this set of 55 point arrays is complete and that there are no additional point arrays as there are no further symmetry axes or displaced vertices to intersect.

For ease of notation and formation of double point arrays, we rescale each of the base polyhedra so that their translation lengths, λ=1, for each of the respective translation vectors, ${\vec{T}}_{2}$, ${\vec{T}}_{3}$ or ${\vec{T}}_{5}$. This rescaling does not change the point arrays nor affect any aspect of their applications to virology. All scaling lengths can be written in the form $2^{m}\phi^{n}$, where m,n=−3,…,3, where $\phi =\frac{1+\sqrt{5}}{2}$ is the golden ratio and $\phi^{\prime }=-\frac{1}{\phi }$, see Table 2. We label these point arrays as $\alpha \;{\left(\mathrm{ICO},\mathrm{DOD}\mathrm{or}\mathrm{IDD}\right)}_{2,3\mathrm{or}5}$ where α=1/λ is the base point array scaling length, e.g.,
$$\underset{\mathrm{base}}{\underbrace{\mathrm{ICO}}}\cup \underset{\mathrm{point}\mathrm{cloud}}{\underbrace{I(\mathrm{ICO}+\;\;\;\;\;\;\;\;\;\overset{\mathrm{Affine Extension}}{\overbrace{\phi {\vec{T}}_{5}}})}}\to \phi^{\prime }\mathrm{ICO}\cup I\left(\phi^{\prime }\mathrm{ICO}+{\vec{T}}_{5}\right)\equiv \phi {\mathrm{ICO}}_{5}$$

Two or more point arrays with the same affine extension vector may be combined to form a larger point array, e.g., $\phi {\mathrm{ICO}}_{5}$ and ${\mathrm{DOD}}_{5}$ may be combined as $\phi {\mathrm{ICO}}_{5}\cup {\mathrm{DOD}}_{5}$, as seen in Figure 6 and the radial distribution of these points are shown in Figure 7a. One interpretation of these combined point arrays, is that the base array is a union of two or more icosahedral polyhedra, e.g., start with the base ϕICO∪DOD and then apply the ${\vec{T}}_{5}$ translation and re-apply icosahedral symmetry. Forming larger point arrays can be desirable for several reasons, including searching for additional constraints on the virus architecture or to find compatible internal or external structural modifications sites. We consider the set of 55 single base point arrays and 514 double base point arrays for our fitness measure.

### 2.3. Major Features of Point Arrays

At first glance, the library of point arrays appears to have a large number of degrees of freedom, which might seem to accommodate any architecture; however the constraints implied by each point array are quite specific, and the restriction of outer features of the viral capsid being coincident with gauge points considerably lowers the freedom of building spherical viruses. An important feature of each of these point arrays is that icosahedral symmetry now occurs on several radial levels (Figure 5) and (Figure 7). Each point array has only one free parameter, the overall radial scaling. Every other aspect is fixed, including the radial distribution and relative angular locations of the points.

The major features of the point arrays are

#### 2.3.1. 55 Unique Single Point Arrays

There are 13 point arrays formed from ${\vec{T}}_{5}=\left[0,1,\phi \right]$ extensions, 17 point arrays formed from ${\vec{T}}_{3}=\left[1/\phi ,0,\phi \right]$ extensions and 25 point arrays formed from ${\vec{T}}_{2}=\left[0,0,\phi \right]$ extensions. Forty-one of these point arrays have elements on the icosahedral symmetry axes at 2 or more radii. The remaining 14 are formed only from DOD and IDD bases with ${\vec{T}}_{2}$ or ${\vec{T}}_{3}$ extensions, and only have one radial level with points on the icosahedral symmetry axes. It is worth noting that in all the affine extensions, no non-icosahedral symmetry appears nor do any new accidental icosahedral axes. This is due to the fact that the icosahedral group is the largest discrete compact rotation group in 3 dimensions. Generally, point arrays involving IDD or ${\vec{T}}_{2}$ extensions have the greatest number of points.

#### 2.3.2. Gauge Points

The set of outermost points from all of the 55 point arrays reveal that there are only 21 unique locations within the AU, known as gauge points because they determine the scaling of the entire point array (Figure 1). Spherical viruses arrange their protruding features to coincide with these points [4]. This limited set of locations has important implications connecting the protrusions of viruses to their internal structure. Once the gauge points are known, there is an absolute maximum distance between the gauge point and the next radial level which cannot be transformed or scaled away, and this next radial level must coincide with the capsid proteins or the entire point array must be discarded. This condition provides new rules for modifying external and interior surfaces of virus capsids. Protruding features located on these gauge points are referred to as Wilson protrusions [4]. The reason there are only 21 locations is that multiple point arrays generate the same gauge points, e.g., $\phi^{\prime }{\mathrm{ICO}}_{3}$ and $\phi DOD_{5}$ begin as different polyhedra, but after translation, each has the same gauge point, see Table 2. The 18 gauge points which are not on symmetry axes originate from exactly two point arrays, referred to as *sister point arrays*.

#### 2.3.3. Sister Point Arrays

There are 26 pairs of point arrays which have identical point clouds and only differ at a single radial level (their base polyhedron). These pairs are referred to as sister point arrays and make up 52 of the 55 single base arrays, see Table 2. We write sister point arrays with a tilde ∼, e.g., $\phi {\mathrm{ICO}}_{2}∼\phi^{\prime }{\mathrm{IDD}}_{5}$, where $\phi {\mathrm{ICO}}_{2}=\phi \mathrm{ICO}\cup I\left(\phi \mathrm{ICO}+T2\right)$ and $\phi^{\prime }{\mathrm{IDD}}_{5}=\frac{1}{\phi }\mathrm{IDD}\cup I\left(\frac{1}{\phi }\mathrm{IDD}+{\vec{T}}_{5}\right)$. Each of these two point arrays have GP:21 and identical point clouds (Figure 8). While sister point arrays are almost identical in their arrangement of points, their combinations of double point arrays are quite different. Two of the sister point arrays are equivalent, up to an overall radial scaling, $\phi {\mathrm{IDD}}_{2}\equiv \phi^{\prime }{\mathrm{IDD}}_{2}$, though when combined to form double point arrays, they are no longer equivalent. There are six enveloping shapes for all single base point arrays ${\mathrm{ICO}}_{3}∼{\mathrm{DOD}}_{3}$, ${\mathrm{ICO}}_{2}∼{\mathrm{IDD}}_{5}$, ${\mathrm{DOD}}_{2}∼{\mathrm{IDD}}_{3}$, ${\mathrm{ICO}}_{5}$, ${\mathrm{DOD}}_{3}$ and ${\mathrm{IDD}}_{2}$.

#### 2.3.4. Double Base Point Arrays

There are 514 unique possible combinations of 2 single base point arrays resulting from the pairing of the 55 starting configurations with the same translation vector [1,2,15] (Figure 6 and Figure 7). In this work, we only consider the 569 single and double base point arrays. We write the largest relative radius point cloud first. We will see that double point arrays can improve the fit and/or suggest sites for modification in the results for the Hepatitis E VLP. In general, double point arrays do not have sister double point arrays, as the sister arrays of each often have different translation vectors. When double point arrays do have sister point arrays which can still be combined, the radial ordering of these point arrays reverses, creating a very different structure than the original double array. For example, $2\phi^{\prime }{\mathrm{IDD}}_{5}\cup {\mathrm{IDD}}_{5}$ has as its combined sisters point array, ${\mathrm{ICO}}_{2}\cup 0.5\phi {\mathrm{ICO}}_{2}$, which swaps the top and bottom point arrays. Double point arrays can be formed by pairing any of the two point arrays in the ${\vec{T}}_{5},{\vec{T}}_{3}$ or ${\vec{T}}_{2}$ list, see Table 2.

#### 2.3.5. Single Free Parameter

While there are many point arrays to consider, it is important to stress that each point array has only a single free parameter, the overall radial scale. We are not free to eliminate points that we do not like, although we may disregard layers of points below the viral capsid when matching point arrays to the capsids, unless the genetic material is known (see fitness procedure in the next section).

### 2.4. Radially Ordered Single Base Point Arrays

We present a new radial ordering of the 55 single point arrays and report their sister point arrays in Table 2. The point arrays have been ordered from largest to smallest relative radius to aid in the construction of similar radially sized double point arrays. For example, finding potential surface modifications for a virus fitted by $\phi {\mathrm{DOD}}_{5}$ would begin by considering $2{\mathrm{IDD}}_{5}$ which provides new geometric constraints which are above those in $\phi {\mathrm{DOD}}_{5}$. These arrays and their 514 double point array combinations will be used to classify viral capsids in this paper.

### 2.5. Point Array Fitness Algorithm

We now present our modified point array fitness algorithm which requires gauge point agreement and low relative surface RMSD (Figure 9).

#### 2.5.1. Identify Protruding Features of a Virus

Our algorithm determines the most radially distal protruding features of each viral capsid. These external features can be critically important in the lifecycle of the virus. In practice, we have found that the outer 3% of atoms by radius in a capsid are sufficient for locating protruding features. We then cluster the atoms in the outer 3% with those beneath them and in the upper half of the capsid. We consider this cluster as our protrusion. We then find the geometric center of mass, treating all atoms as if they have the same mass. This process is described in more detail in our previous work on protruding features [4].

#### 2.5.2. Determine Gauge Point Scaling

We start the 21 gauge points 5 angstroms beyond the atom with the largest radius. We then allow them to fall radially onto the capsid. A gauge point is stopped when it reaches its nearest distance to the protein surface, which we model as an overlap of 1.5 Å spheres which is slightly smaller than the average van der Waals radii of each of the heavy atoms [22]. If a gauge point would fall through the surface of the capsid, we stop it at its point of closest approach. We only consider point arrays with gauge points located on or near these protruding features, referred to as admissible gauge points [4]. This dramatically reduces the number of point arrays which need to be considered when characterizing a virus, see Table 2. The gauge points are presented in Figure 1. An example of this step applied to Hepatitis B can be seen in Figure 10. The yellow gauge points on the 3-fold axis would have fallen through the capsid, but were stopped at the location of nearest approach. The determination of the admissible gauge points will prove to be a critical step in understanding the restrictions conferred by point array geometries on the entire capsid structure.

#### 2.5.3. Scale and Truncate Point Arrays

Next we scale point arrays with admissible gauge points to match the virus capsids. Generally, point arrays cover a larger radial extent than the capsid proteins, going well into the interior of the virus. We interpret these interior points as characterizing the constraints on genetic material contained within and therefore truncate them from the point arrays. We carefully truncate the point arrays as dropping interior points while optimizing scaling can dramatically alter the RMSD fit. When a point array is first scaled to gauge point, we determine the nearest point array elements to all the capsid atoms and keep any points above the minimum radii of this set. Next we compute the distance to each point below the protein capsid’s inner surface and keep all those points within 4 Å. This set of point array elements is referred to as our truncated point array. We find that most capsids use a truncated point array.

#### 2.5.4. Compute RMSD from Truncated Point Arrays to the Viral Capsid Proteins

The surface RMSD for the point array elements within the asymmetric unit according to [2]
(2)$$R_{\mathrm{surf}}={\left(\frac{\sum_{i=1}^{N}m_{i}p_{i}d_{i}^{2}}{\sum_{i=1}^{N}m_{i}p_{i}}\right)}^{1/2}$$
where $d_{i}$ is the minimum distance from the *i*th point array element to the nearest protein surface(s), $p_{i}$ is the protein multiplicity (number of distinct proteins) near the point *i* (e.g., 5 for a point on a 5-fold axes or 2 if two proteins are equidistant from the same point). Finally, $m_{i}$ is the number of times the point appears in the full point array (e.g., 12, 20, 30 etc.) and *N* is the total number of point array elements. Protein multiplicity $p_{i}$ is a weighting factor in the RMSD for when two or more proteins are roughly equidistant from the same point to within a few tenths of an angstrom, e.g., if a point array element were sitting on a 3-fold axis, it would be counted 3 times. Figure 10 provides an examples of protein multiplicity and the number of times a point appears in the AU of the point array $m_{i}$.

We minimize the RMSD by radially shunting the entire point array radially by ±5 Å in 0.25 Å steps. During this shunting, we eliminate point arrays which contain points within the protein surfaces or small pockets, as point arrays represent external material boundaries [2,15]. We do not allow further interior point array elements to be cutoff during this optimization, we refer to this as gauge fixing. This modified algorithm eliminates most spurious point arrays from considerations, improving upon the fits of [2].

#### 2.5.5. Determine Best Fit Point Arrays

In general, we only find a handful of point arrays which have the correct gauge point(s) and low RMSD. Often a point array and its sister point array will both meet the same criteria and without knowledge of the genetic material, they are indistinguishable. The following criteria to break ties
If a point array has a lower RMSD score by 0.5 Å or more.Have at least one element near each protein.Encase the protein capsid with points above and below.Have a better agreement with the gauge point fits, as seen in Figure 11.Have more points of contact with capsid proteins, e.g., each point on the five-fold axes have at least 5 points of contact with protein surfaces. We consider this step after checking gauge point fits (d), as the number of contacts can be quite large for point arrays with IDD bases or ${\vec{T}}_{2}$ extensions which can considerably lower the RMSD score.Have the largest range of prevalence (Figure 10), indicating a wide range of radial agreement [2].
The most important discriminant in this process is step 1, determination of the protruding features and the admissible gauge points. Afterwards when point arrays have comparable RMSD values, we consider point arrays which pass criteria (b) and (c) to be the best fitting point arrays. If a tie still exists, we consider steps (d)–(f) in order, though most point array fits are determined by step (c). When ties still exist, we report both point arrays, which is a common occurrence with sister point arrays, which we consider to be equally valid (indistinguishable) (Figure 9). These ties might further be broken by including genetic material where available. The results of this algorithm for the viruses studied in this paper can be found in Table 3.

### 2.6. Comparison with Previous Measure

We will now review the key differences between the fitness measure of Keef et al. [2] and our own measure.

#### 2.6.1. Gauge Point Agreement

We only allow point arrays with gauge points near the protruding features of viruses, dramatically limiting the number of point arrays to consider. This set of criteria is based on angular proximity with the protruding features, rather than overall scale of the point arrays, which has provided the single largest improvement to the fitness criteria.

#### 2.6.2. Simplified RMSD Measure

The prior fitness measure used a topological RMSD in addition to the surface RMSD, via $\mathrm{RMSD}={\left({\mathrm{RMSD}}_{\mathrm{surf}}^{2}+{\mathrm{RMSD}}_{\mathrm{top}}^{2}\right)}^{1/2}$, which was very sensitive to the overall radial scaling of the point arrays, and could allow a poor ${\mathrm{RMSD}}_{\mathrm{top}}$ score to be overcome by an abundance of good surface points deep within the capsid. These point arrays often included IDD and ${\vec{T}}_{2}$ elements, failed to describe the outer surface of the virus. By contrast, gauge point agreement is independent of radial scaling.

#### 2.6.3. Gauge Fixing of Truncated Point Arrays

We make a single interior point array cutoff when the initial gauge point scaling is determined, which prevents poor interior points from being accidentally removed while optimizing the ${\mathrm{RMSD}}_{\mathrm{surf}}$ score.

#### 2.6.4. Recognition of Sister Point Arrays

Many point array fits appear distinct, but upon inspection are doubly represented at the level of single base point arrays due to the radial truncation of the point arrays.

#### 2.6.5. Tie Breaking Criteria

We introduce several criteria to break any RMSD ties, such as requiring at least one point per protein (Figure 9). We also added a new criterion, that all point arrays have at least one point which corresponds to each protein within the asymmetric unit (AU). While we consider prevalence, we generally find that excluding point arrays due to their being located within a small pocket of a protein to be sufficient.

## 3. Results and Discussion

The radial and angular locations specified by point arrays serve as contact points for proteins, as well as external and internal material boundaries. The majority of points are located on the icosahedral great circles. See Figure 5, Figure 7 and Figure 8 for examples of radial distributions of point arrays. We interpret these locations as boundary constraints in the sense of Janner’s encasing forms [5,6,7,8,9], for the protein capsid and genetic material contained within.

This interpretation makes sense, as proteins cannot be located directly atop the symmetry axes, so these points provide a convenient location for the proteins to maintain contact while adhering to the overall symmetry. We believe that some of these points might act as mechanical equilibrium locations for the icosahedral vibrational modes of the capsids. Given that multiple proteins often make contacts at these points and that meeting these geometric restrictions appears to be advantageous, it is plausible that proteins could be oscillating around these locations similar to a mass on a spring. We will see below that the chemically identical protein dimers of HBV have different vibrational properties which are consistent with the locations of the point array elements.

While it remains unclear exactly why viruses are adopting these geometries, we agree with the conjecture that viruses adhering to this extended symmetry could help explain their inherent stability [2]. We suspect that the structural rearrangements of capsid proteins throughout maturation can be explained by the capsid transitioning from one point array to another, as we will see with CCMV below. A capsid could in principle gain or lose stability by changing point arrays. Maturation can be triggered by relatively small, local modifications to proteins that could have a dramatic effect on the RMSD fit. These point array locations serve to impose new geometric constraints on the spatial arrangement of protein chains, which can serve as a pseudo-steric hinderance to adding new ligands. These constraints can indicate where modifications are likely to succeed, although not what form they should take. We will see that in MS2, the gauge points reveal critical structural locations which are nearly impossible to modify [23,24].

It can often be frustratingly difficult to modify protein capsids at particular sites. The exact reasons are often unknown as these sites appear ideal based on local chemistry, solvent accessibility and a lack of steric hinderance. As we will see, knowledge of point array fits provides insight for these modifications, based purely geometric constraints that would be overlooked in other analyses. The stability implied by a virus capsid’s adherence to the geometric constraints of point array might be disrupted by deviations from these constraints. It is also possible that these point arrays could specify multiple sites which must be modified to remain stable, depending on the distribution of point arrays by radius and the rules of combining point arrays.

### 3.1. Virus Point Array Classification

Here we present our results of the Point Array Fitness Algorithm for 16 viruses ranging over *T* = 1, 3, 4 and 7 virus capsids of RNA and DNA viruses in Table 3. Overall, we found that all these viruses had only a single best array, up to sister array fits, with RMSD values generally less than 2 Å. The criteria which made the largest difference after RMSD ties, were requiring a description of all proteins, best gauge point agreement and points above and below the capsid. We found that the only fits which could not be distinguished from other point arrays were those which resulted from sister point arrays. We generally find that relatively thin viral capsids, such as GA and MS2, are equally co-fit by their sister point arrays. This brings the effective number of distinct single base point clouds down to 29 when considering thin capsids. Without knowing the locations of the genetic material, it is not yet possible to distinguish between these two point arrays. The three *T* = 1 capsids in Figure 2, Infectious Bursal Virus, Hepatitis E VLP and the Penton Base of Adenovirus A (Ad3) do not have ties for their best fit.

We see from Table 3 that knowing the *T*-number does not predict which point array a virus will utilize, nor does knowledge of the point array predict the *T*-number. Each of these structural classifications communicates different information about the capsid. We have not yet found any viruses with a 3-fold protrusion in the Viper Database, which is consistent with our earlier work [4], though there are several viruses with protrusions near GP:5 and 7. Based on these results, we suspect that most viral capsid will have only a single best fit point array, up to its sister array. We also acknowledge that if a virus simultaneously conformed to several point array geometries, it could be found to be more stable.

The only virus which we needed to consider prevalence for tie breaking was STMV (T=1, 1a34 [26]), which had two fits $\phi^{\prime }{\mathrm{DOD}}_{5}$ and ${\mathrm{ICO}}_{3}∼{\mathrm{DOD}}_{5}$, with RMSD values of 0.8 Å and 1.2 Å, respectively. However this turned out to be coincidental, as one of the interior points was essentially stuck within a protein van der Waals pocket, and only permitted the point array to be scaled up or down by 0.25 Å, less than the diameter of a hydrogen atom. We therefore eliminated the point array $\phi^{\prime }{\mathrm{DOD}}_{5}$ from consideration. Of the 16 viruses presented here, each affine extension, 2-fold, 3-fold and 5-fold were used. We also found that 5 of the 6 possible enveloping point array shapes were needed to describe these viruses with only ${\mathrm{DOD}}_{3}$ like capsids being absent, consistent with our lack of 3-fold protrusions [4].

### 3.2. Advantage of Sister Point Arrays

When a virus is equally fitted by sister point arrays, there may be several possible double base point array combinations which indicate modifications open to the virus. For example consider STMV, which is equally well fit by ${\mathrm{ICO}}_{3}$ and ${\mathrm{DOD}}_{5}$. From Table 2 we see that ${\mathrm{ICO}}_{3}$ can be combined with ${\mathrm{IDD}}_{3}$, ${\mathrm{DOD}}_{3}$ and ${\mathrm{IDD}}_{3}$ and that ${\mathrm{DOD}}_{5}$ can be combined with ${\mathrm{IDD}}_{5}$, $2\phi^{\prime }{\mathrm{IDD}}_{5}$ and ${\mathrm{ICO}}_{5}$ allowing potentially 6 distinct surface modifications to this *T* = 1 virus. For more examples of combining point arrays, see Figure 7.

### 3.3. Penton Base of Adenovirus Ad3 Dodecahedron (HEV, T = 1, 4aqq)

The first viral system that we explore is human adenovirus serotype 3 (Ad3) [20]. During its lifecycle an excess of free pentameric Ad3 capsid proteins are synthesized compared to its trimeric fibers. These capsid proteins form a 12-component dodecahedral virus-like particle (Figure 12) which is stabilized by strand-swapping near the N-termini. The complex of these dodecamer capsids with the trimeric fiber proteins are responsible for virus penetration into the cell [20]. We find remarkable agreement between the precise location of the strand-swapping and the geometric constraints of the point arrays. The strand-swapping is required for stabilization of the capsid [20] and we have no ability to remove this geometric constraint from the overall constraint set, demonstrating how powerful the lack of free parameters is in the use of point arrays. Adenovirus Ad2, a different serotype, has residues which stabilize the dodecamerization [38] and does not use the strand-swapping.

We find that this capsid is best fitted by ${\mathrm{IDD}}_{5}$ and it is not equally co-fit by its sister point array ${\mathrm{ICO}}_{2}$. While these two point arrays contain nearly identical points, the radial location of the IDD and ICO bases (Figure 8) lead to a considerable difference in their RMSD see Table 3. The difference comes from a hole in the pentameric protein volume (Figure 12c), which leaves the inner protein surface several angstroms away from the 5-fold point in ${\mathrm{ICO}}_{2}$. This result is distinct from most other fits, as it employs all the radial levels of the point array without requiring any genetic cargo within. This RMSD fit is not perfect however, as there are points in between the pentameric towers which are several angstroms off the protein surface between the 3-fold and 5-fold axes leading to an RMSD of 3.7 Å.

### 3.4. Hepatitis E VLP (HEV, T = 1, 3hag)

The Hepatitis E virus-like particle illustrates how combining point arrays can lead to an improved point array description (Figure 13). Of the 55 single point arrays, HEV is best fitted by $2\phi {\mathrm{IDD}}_{2}$. However, there are no points located on the interior surface of this capsid. When we consider the additional 514 double point arrays, we find that $2\phi {\mathrm{IDD}}_{2}\cup \phi^{2}{\mathrm{IDD}}_{2}$ lowers the RMSD from 3.2 Å to 2.8 Å and provides points on the interior of the capsid as well. The fit of $2\phi {\mathrm{IDD}}_{2}$ is interesting, because it is the largest radius point array in the ${\vec{T}}_{2}$ extensions (Table 2). We believe that this implies it will be difficult to add new surface features beyond the radius of 2-fold Gauge Point 15 without also changing some interior structure of the capsid to fit the new embedding array. This lack of external modifications is consistent with the *T* = 1 HEV being non-enveloped. There may be another way to modify *T* = 1 HEV, by modifying the capsid to fit the sister point array $0.5\phi^{\prime }{\mathrm{IDD}}_{2}$ instead. This approach however introduces new challenges as this point array is now the smallest radius point array within the ${\vec{T}}_{2}$ extensions, and thus there would no longer have any interior point arrays which to make contact with the inner surface, which could indicate a lack of stability.

As it can be challenging to fully grasp how well point arrays fit all the nooks and crannies of the van der Waals surface of the protein capsid, as well as the interior point of contact, we will now examine a ribbon side view around the 2-fold axis of HEV in Figure 14. Here it is easier to see how the points fit in and around the boundaries of each protein, providing sites of coordination for their contacts. We see that the protruding dimer perfectly fits around the 2-fold GP: 15 and that this contact occurs again at points along the 5-2 great circle and 2-fold axes inside the capsid surface. The bulk points also provide a set of geometric constraints, to which the individual proteins are clearly adhering. In future work, we will explore the native *T* = 3 HEV.

### 3.5. Bacteriophage MS2 (MS2, T = 3, 2ms2)

The protein capsid of Bacteriophage MS2 fits entirely in between the outer two radial levels of the sister point arrays $\phi {\mathrm{ICO}}_{3}∼\phi^{\prime }{\mathrm{DOD}}_{5}$, which each have five radial levels total. Knowledge of the interior structure of the ssRNA should enable us to distinguish between these two point arrays. Recently a series of very clever experiments were conducted by Hartman et al. [23,24], where they systemically replaced every amino acid, one by one, of the subunit proteins of MS2 with each of the 19 other possible α-amino acids. As MS2 encapsulates its RNA genome upon formation of the capsid, they were able to determine which locations they could swap out and still form a stable virus. Curiously, they found that they had almost no ability to replace the amino acids in the pentamer and hexamer protrusions, which are in identical locations in each of the three proteins of the AU. We see in Figure 15 that the gauge points sit on the sides pentameric protrusions, where they were almost completely unable to make any amino acid substitutions. They also had only partial success in substitutions on the opposite side of the same protrusion with nearly equivalent amino acids. This suggests that this geometric location is of critical importance to the stability of the capsid. This immutability is unexpected, as the protruding feature is a relatively small protein loop on top of a much larger capsid, and is not involved in enclosing the genome. While the location and composition of these surface loops are important with regards to host infections, it should still be physically possible to make stable, even if not viable, mutations. We believe that this is clear evidence that viruses are indeed gaining stability by adhering to the geometric constraints of point arrays, and that most mutations or changes in this loop on the pentamer would decrease the overall stability, as it would deviate from the point array. They also found that they were able to freely modify the f-g loop (Figure 15), which is a desirable location as it is not sterically hindered and should easily accommodate different amino acid residues. This result is also in agreement with the point array description, as it does not impose any restrictions near the center of the hexamer, rather only on the interior surface, several residues away from the f-g loop. Their results also suggest it is difficult to swap amino acids near the inner points, though this difficulty is not unexpected as it is located near protein β-sheets.

The capsid structure of the bacteriophage GA (*T* = 3, 1gav [28]) is nearly indistinguishable from the bacteriophage MS2. We also find this to also be true in terms of the point array descriptions, each being well fit by $\phi {\mathrm{ICO}}_{3}∼\phi^{\prime }{\mathrm{DOD}}_{5}$ with RMSD 0.7 Å for MS2 and 0.2 Å for GA. In our previous study of protruding features [4], we observed that the orientation of the hexamer surface loops of bacteriophage GA were parallel to the surface, rather than oriented radially as with MS2. This variance is perfectly consistent with the point array description; while the protruding features of the pentamers and hexamers are chemically identical, their geometric constraints are not as there are no restriction on the hexamer loop. The point array description indicates that it should be possible to selectively modify the hexamer protrusions on GA and MS2 post-assembly.

### 3.6. Hepatitis B (HBV, T = 4, 1qgt)

The Hepatitis B (*T* = 4) capsid is composed of 120 protein homodimers and is best fitted by the double base point array ${\mathrm{ICO}}_{2}\cup 0.5{\mathrm{ICO}}_{2}$ with has an RMSD of 1.3 Å. There are two distinct dimeric units, the AB and CD dimers, which have the same primary and secondary structure, though differ slightly in their tertiary structures when part of the capsid (Figure 16). Each of these dimeric units are sandwiched between elements of the point array (Figure 16b,c). The geometric constraints on the AB and CD dimers are not equivalent as the AB dimer is restricted to meet at the top of the α-helices and the CD dimer near the center of the α-helices interface. If we treat the elements of the point arrays as equilibrium points for thermal fluctuations of the capsid, we would expect the dimers to have different flexibility parameters. We would expect the AB dimer to be stiffer than the CD dimer, as it has fewer fluctuations which leave it in contact with the gauge point at the top of the dimer and the point below (Figure 16b). In contrast, the CD dimer has more fluctuations available which leave the α-helices in contact along their interface, and the points below (Figure 16c). This difference was indeed observe by Hadden et al. [39] in their enormous all-atom, non-symmetry enforced, molecular dynamics simulation of HBV over a timescale of 1 μs with 2 fs time steps. They found that the CD dimers were more flexible than the AB dimers. Their simulation also found that the capsid was never truly converged to an icosahedral capsid despite the long relaxation time, which could imply that the virus only has icosahedral symmetry on average. Another interesting feature of the HBV (*T* = 4) capsid is that the hexamers are centered on 2-fold axes, which is consistent with *T* = 4 geometries. This requires that the hexamers have 2-fold rather than 3-fold symmetry. The hexamers accomplish this by arranging the CD dimers and AB dimers differently, creating a squashed 2-fold symmetric hexamer (Figure 16d).

The all-atom simulation by Hadden et al. [39] also demonstrated that HBV was never found to be in an idealized icosahedral configuration. Wang et al. [40] have suggested that spherical viruses may only be icosahedral on average, and that it might be incorrect to characterize deviations from icosahedral symmetry as defects rather than the norm. This fact would not be in in conflict with point arrays, as they provide a network of radially distributed locations which can facilitate multiple protein contacts. These contacts could serve to stabilize the capsids in lieu of perfect icosahedral symmetry. In this view, the points would specify equilibrium locations, or possibly nodes, for the proteins to oscillate about due to thermal fluctuations or other defects. It is important to point out that while these arrays provide many new constraints, asymmetric distortions are certainly possible as the arrays do not fully constrain the capsid. It has also suggested that reactive sites may be the reasons for these defects [40]. We are currently seeking data on these systems to test out point arrays.

### 3.7. Cowpea Chlorotic Mottle Virus Maturation (CCMV, T = 3, 1cwp)

CCMV is a well-known *T* = 3, ssRNA virus with a well-studied pH induced maturation. One of the first applications of point arrays was to study possible maturation pathways in higher-dimensional space to use quasi-crystallographic techniques to characterize the maturation [41]. We begin by analyzing the native state which is best fitted by $\phi^{\prime 2}{\mathrm{ICO}}_{3}$ with an RMSD of 0.7 Å and Gauge Point 5. We excluded one of the point arrays $\left(2\phi^{\prime }{\mathrm{IDD}}_{3}\right)$ considered by [41] as it had a point located within a protein pocket. The hexamer subunits seem to be the most important to the stability of the capsid in terms of point arrays. The hexamer subunit is sandwiched above, below and at the two-fold axes by point array elements (Figure 17). While about 200 atoms of the RNA structure are known in the native capsid, we did not consider them in our point array fit as we do not know any of the RNA structure in the mature capsid. The best fit point array $\phi^{\prime 2}{\mathrm{ICO}}_{3}$ extends about 50 Å below the interior surface of the capsid, so there may also be restrictions placed upon the interior ssRNA.

The all-atom coordinates of the swollen CCMV capsid are not known from a crystallographic data, rather they come from fitting the native capsid protein structures into cryo-EM data [25]. We found several apparent point array fits, which are given in Table 4. Interestingly, the native state point array still fits the capsid after maturation, though not as well, as the RMSD increased by 1.2 Å This lower RMSD agreement could signal the loss of stability of the capsid which is observed after maturation [25]. In this case, we consider the sister point array to be a better description, as the interior center of the hexamer contracts relative to the native state to meet a new 3-fold point array location and therefore has 120 more points of contact with the proteins (Figure 18). This demonstrates how point arrays can be a powerful tool for studying maturation, while the *T*-number remains constant throughout, protein orientations are able to change considerably.

Numeric studies of the normal modes of the protein capsid showed that for CCMV, the dominant change in structure during maturation could be captured through the breathing mode, or roughly uniform expansion of the protein components [42]. Given that the mature state is still well characterized by the native state array, though with a larger radius, this is consistent with uniform radial expansion. One aspect which is not captured in their numeric studies however is the changing shape of the interior of the hexamer which now makes the sister point array viable (Figure 18d).

### 3.8. Cowpea Mosaic Virus (CPMV, pT3, 1ny7) Lysine Analysis

CPMV is a pseudo-*T*=3 capsid composed of 60 multi-domain proteins which mimics a *T* = 3 capsid geometry with 12 pentamers and 20 hexamers. CPMV is a robust icosahedral capsid, with many successful surface modifications [17]. The surface reactivity was carefully studied site by site through systematically replacing the 5 solvent exposed lysine residues one at a time with arginine residues [32]. There are two nearly identical point arrays which are good fits to CPMV, $\phi^{\prime }{\mathrm{ICO}}_{5}\cup 2\phi^{\prime 2}{\mathrm{IDD}}_{5}$ and $\phi {\mathrm{ICO}}_{5}\cup \phi {\mathrm{DOD}}_{5}$, which after being truncated only differ by a single point (Figure 19). While we typically consider the point array with more points of contact to be a better fit, it does not mean that other point arrays simply discarded. As we will see, CPMV likely switches point arrays after the lysine attachments (Figure 19). For our analysis, we treat the 3 protein domains A, B and C as separate protein chains. Our comparison of the point arrays fit and their restrictions to the relative reactivity data is given in Table 5.

### 3.9. Bacteriophage HK97 Prohead II (HK97, T = 7l, 3e8k)

For our final analysis, we examine dsDNA Bacteriophage HK97 Prohead II (HK97, T=7l, 3e8k [37]), one of the few procapsid X-ray structures known. This viral capsid has a unique protein motif, referred to as chainmail [37,43]. The best fit point array places geometric restrictions on the hexamers which break local 3-fold symmetry (Figure 20). It has been observed that there is a conserved subunit interaction near the three-fold axes [37], which point arrays might be able to explain. The best fit point array was $\phi^{\prime }{\mathrm{ICO}}_{5}\cup \phi^{\prime }{\mathrm{IDD}}_{5}$ (Figure 20). There were two other point arrays also with close fits, $\phi {\mathrm{ICO}}_{5}\cup \phi {\mathrm{DOD}}_{5}$ with RMSD 1.9 Å and ${\mathrm{ICO}}_{5}\cup {\mathrm{DOD}}_{5}$ with RMSD 2.1 Å, though these failed to have any points on the interior of the capsid surface. Interestingly, none of these three point arrays described all proteins; however, all proteins were described by the three total point arrays. Unfortunately, we are unable to simply merge these point arrays as they have different scaling lengths and would have to shift relative to each other (Figure 6). It is possible that the capsid is stabilizing itself by adopting multiple point arrays until it is packaged with DNA. In future work we will explore how the point array description changes throughout the maturation of HK97.

### 3.10. Limitations of Point Arrays

As with any new idea in science, we must step back and question these results. As of now, we do not offer a deterministic understanding of precisely why viruses use these point arrays, though we posit that an explanation involving energetics and stability must exist. Even if it were shown that these point arrays are simply a geometric coincidence of icosahedral packing, rather than energetically favorable, these results would still offer valuable insights into the best ways to determine where and how to modify viral capsids. As we have seen these point arrays describe a range of Triangulation numbers, though we do not believe it will hold for all spherical viruses. As the diameter of viral capsids increases, the thickness of their capsids remains relatively constant [44] whereas the spacing between radial levels increases with capsid diameter. At some point, the capsids will no longer be able to meet the radially distributed requirements and at most will satisfy only the gauge point locations, suggesting another scheme or dramatic modification would be needed for giant spherical viruses.

## 4. Conclusions

Our characterization and understanding of spherical virus architecture has been enhanced using the icosahedral point arrays first introduced by Keef and Twarock [1]. This characterization is independent of Triangulation number and quasi-equivalence. It provides entirely new angular and radial constraints on the relative locations and orientations of the capsid proteins and their interior genetic cargo. Thus, providing a complementary set of geometric restrictions in tandem with the constraints on the pentamer and hexamer subunits, specified by the Triangulation number for spherical viruses.

While viruses could, in principle conform to multiple point arrays, we have shown that over a range of Triangulation numbers from *T* = 1 to *T* = 7 capsids, they are generally characterized, up to sister point arrays, by a singular point array. Our measure clarifies and simplifies the fitting of point arrays from previous work [2,15]. The recognition of sister point arrays aids in our understanding of single point arrays as well as the construction of double point arrays. The geometric constraints combined with a lack of free parameters, make these arrays an exciting predictive tool for understanding viral architecture, including modifications, drug placement and potential limits on viral evolution.

All the viruses that we have investigated conform to the restrictions of the affine extended point arrays. We believe that these points may indicate sites of stabilization, conditions for modifications, as well as mechanical flexibility, which would otherwise be missed by traditional chemical analysis, such as binding affinity, solvent accessibility, steric hinderance and local reactivity. These sites are also not required by Triangulation number nor by quasi-equivalence. We have explained the surprising geometric importance of the immutable surface loops on the MS2 penton, as well as a consistent description of the relative reactivity of surface lysine residues on CPMV. We demonstrated that these geometric locations also correlate with locations of mechanical equilibrium and flexibility, as in the case of the AB and CD dimers of HBV. In addition, these restrictions are important throughout the capsid interior, as seen in the freedom to modify the hexametric subunits known as the f-g loop of MS2, as well as the location of the critically important strand-swapping stabilization of the Adenovirus 3 dodecahedral base.

As point arrays are also able to specify important internal surface locations, they could be useful for genome attachment, interior surface modifications, as well as drug placement for nanomedicine VLPs. Double point arrays may offer a new predictive tool for surface and interior modifications, as we saw with the double point arrays fit of *T* = 1 HEV. This fit also indicated that there was a lack of potential surface modifications without potentially large reorientation of the capsid proteins and/or multiple sites of modification. This also applies to decoration proteins and other ligand attachments. These point arrays capture important structural changes in the maturation of CCMV and may provide similar details for systems like HK97 and P22. Rossmann [45] suggested that viruses conserve tertiary and quaternary structure under mutation, we suggest that this may be due, in part to preserving key geometric relationships of the point arrays. This new understanding may provide new clues to past and future viral evolutionary pathways.

In the future, we will work on developing a web tool which will allow scientists and engineers to have their viruses categorized, key geometric locations indicated and to receive point array suggestions for modifications. We are also exploring the link between point arrays and mechanical flexibility via normal mode analysis. While there is still much to learn about how viruses use these point arrays, it is clear that viruses are playing an intriguing game.

## Figures and Tables

**Figure 1 viruses-12-00467-f001:**
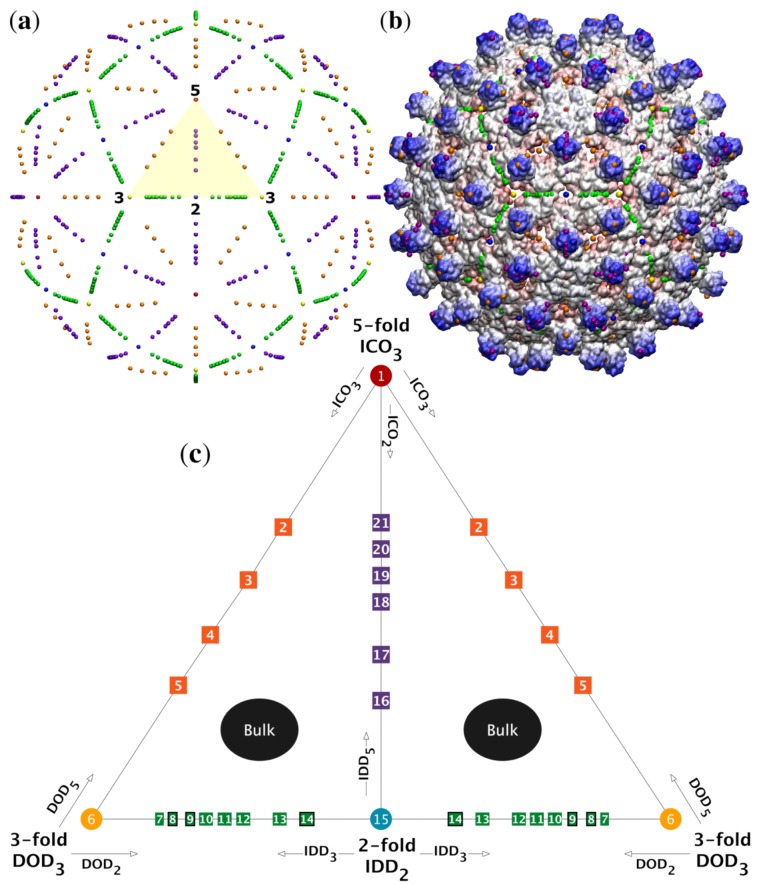
(**a**) The spatial distributions of gauge points have icosahedral symmetry, i.e., they are arranged with 2-, 3- and 5-fold symmetry as shown. The gauge points all lie on the 15 icosahedral great circles which connect nearest neighbor symmetry axes [4] and are colored as in Table 1. We refer to the arcs of the circles subtending the 2-, 3- and 5-fold axes as the 5-2 GC, 5-3 GC and 2-3 GC. The volume bounded between these arcs is known as the asymmetric unit (AU) and is a representative one-sixtieth section of the entire capsid (yellow). (**b**) The gauge points have been placed on a radially colored HBV (1qgt) capsid at their nearest possible distance from the protein surface. We will later see that the only admissible gauge points are the purple and orange elements sitting atop the protruding dimers. (**c**) There are 21 unique gauge points in the AU, see Table 2. The gauge points are formed from 2-, 3- and 5-fold translations of the icosahedron (ICO), Dodecahedron (DOD) and Icosadodecahedron (IDD). The gauge points along the 2-, 3- and 5-fold axes result solely from ${\mathrm{IDD}}_{2}$, ${\mathrm{DOD}}_{3}$ and ${\mathrm{ICO}}_{5}$ point arrays, respectively. The off-axis gauge points, e.g., GP:2-GP:5, each result from only nearly identical point arrays known as sister point arrays. For example, Gauge Point 4 (GP:4) is only created from $\phi^{\prime }{\mathrm{ICO}}_{3}$ and $\phi {\mathrm{DOD}}_{5}$. In total 36 of the 55-point arrays have gauge points off-axis. There are no gauge points in the bulk regions not located on the great circles, however interior points of the point arrays can be located there.

**Figure 2 viruses-12-00467-f002:**
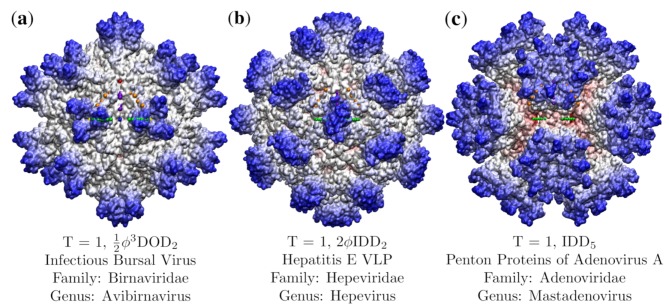
A structural comparison showing the wide diversity in viral capsid shapes of even the simplest T=1 viruses. Each virus capsid has been radially colored, with red being the most interior, and blue the exterior. These three, (**a**) Infectious Bursal Virus (Avibirnavirus, 2gsy [18]), (**b**) Hepatitis E Virus-like particle (Hepevirus, 3hag [19]), and (**c**) Penton Base of Adenovirus A (Mastadenovirus, 4aqq [20]), were chosen as they have almost no overlap in their protruding features. The best fit point array for each capsid is listed. We will see that these point arrays describe unique features and complement the *T*-number classification. The asymmetric unit of each capsid is contained by the triangular region of gauge points (Figure 1).

**Figure 3 viruses-12-00467-f003:**
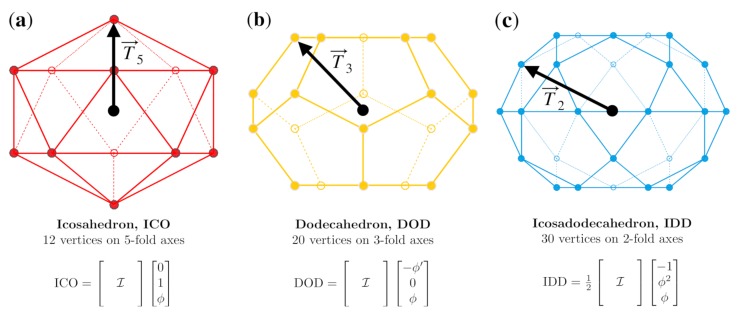
The three standard polyhedra with icosahedral symmetry and the affine extension translation vectors. (**a**) The 12 vertices of the icosahedron are on the six 5-fold axes, and the structure is generated by applying all 60 icosahedral rotations on a single point [0,1,ϕ], which also serves as the translation vector ${\vec{T}}_{5}$. (**b**) The 20 vertices of the dodecahedron are on the ten 3-fold axes, and the structure is generated by applying the full icosahedral group to the point $\left[-\phi^{\prime },0,\phi \right]$, which also serves as the ${\vec{T}}_{3}$ translation vector. (**c**) The 30 vertices of the icosadodecahedron are on the fifteen 2-fold axes, and the structure is generated by applying the full icosahedral group to the point $\frac{1}{2}\left[-1,\phi^{2},\phi \right]$, which serves as the ${\vec{T}}_{2}$ translation vector.

**Figure 4 viruses-12-00467-f004:**
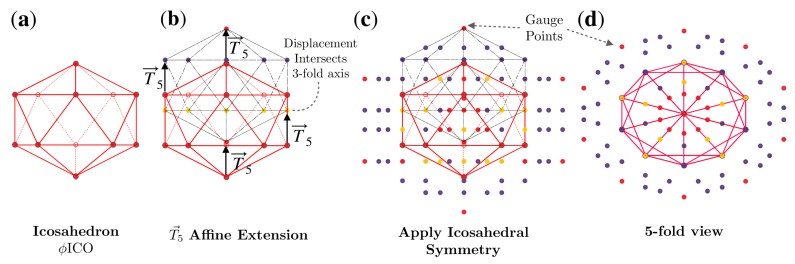
Formation of the $\phi {\mathrm{ICO}}_{5}$ point array [1]. (**a**) The base of this point array is an icosahedron scaled up by multiplying by ϕ. (**b**) The base point array is then translated by ${\vec{T}}_{5}$. Under this ${\vec{T}}_{5}$ extension, the base point array consists of 4 separate levels. The lowest level is a single vertex which remains on the 5-fold axes, the next level of 5 vertices all intersect 3-fold axes (all with the same radius). The next level is located on the 5-2 great circle of the AU, resulting in 60 points. Finally, the gauge point of the point array is created by the largest radius point (Figure 5). (**c**) Icosahedral symmetry is now applied, creating the full point array (base and cloud), where the points are colored as in Figure 1. As the translation was ${\vec{T}}_{5}$, the resulting gauge point remains on the 5-fold axes (Figure 1). (**d**) A look down one of the twelve 5-fold axes shows that the point array has icosahedral symmetry. This entire point is enveloped by a now larger icosahedron, with the purple points sitting on its edges. The formation of point arrays can equivalently be described as placing an icosahedron centered at each of the original base vertices to create the point cloud.

**Figure 5 viruses-12-00467-f005:**
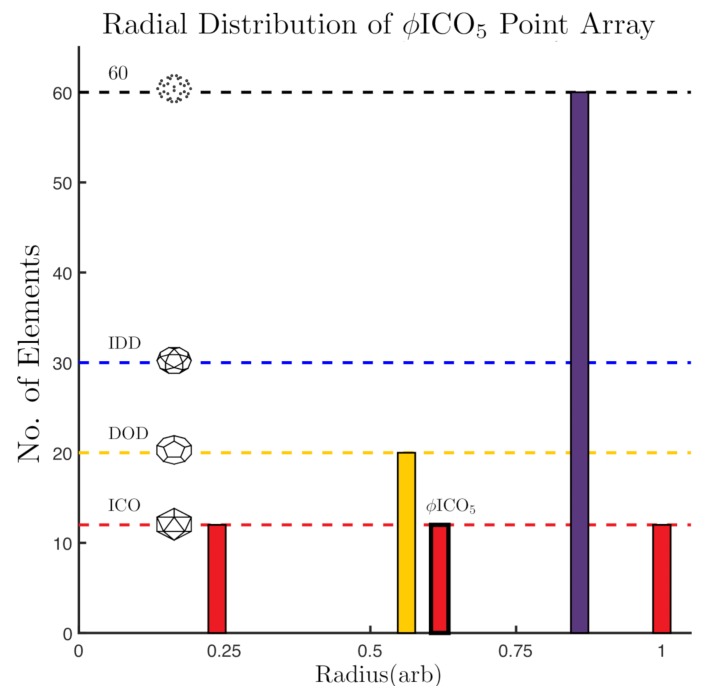
A histogram of the point array elements by radii in arbitrary units. The base array ϕICO has a bold outline around the 12 points at r∼0.64. The gauge points are on the 5-fold axes (Gauge Point 1). In total there are now 3 icosahedrons and one dodecahedron, all with different radii. There is also a layer of 60 points along the 5-2 great circle of the AU; due to the 60 rotations of the icosahedral group, any point not located on a symmetry axes will produce 60 points after the symmetry is applied (Figure 4). The cardinality or size of this point array is 116=12(ICO)+20(DOD)+12(ICO)+60(5−2)+12(ICO).

**Figure 6 viruses-12-00467-f006:**
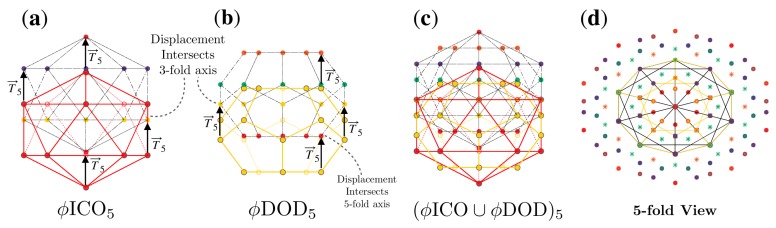
Formation of a double base point array. (**a**) Point array $\phi {\mathrm{ICO}}_{5}$ and (**b**) point array $\phi {\mathrm{DOD}}_{5}$ have the same affine extension ${\vec{T}}_{5}$ and may therefore be combined to form (**c**) ${\left(\mathrm{ICO}\cup \mathrm{DOD}\right)}_{5}$, a double base point array. (**d**) A view down a 5-fold axes after applying icosahedral symmetry, where the points originating from $\phi {\mathrm{DOD}}_{5}$ are shown as asterisks (✳). All that is required to combine these two point arrays is that they have the same translation vector ${\vec{T}}_{5}$ and it is coincidental that both $\phi {\mathrm{ICO}}_{5}$ and $\phi {\mathrm{DOD}}_{5}$ intersect at a 3-fold axes during the affine extension. The full radial distribution of points is shown in Figure 7a.

**Figure 7 viruses-12-00467-f007:**
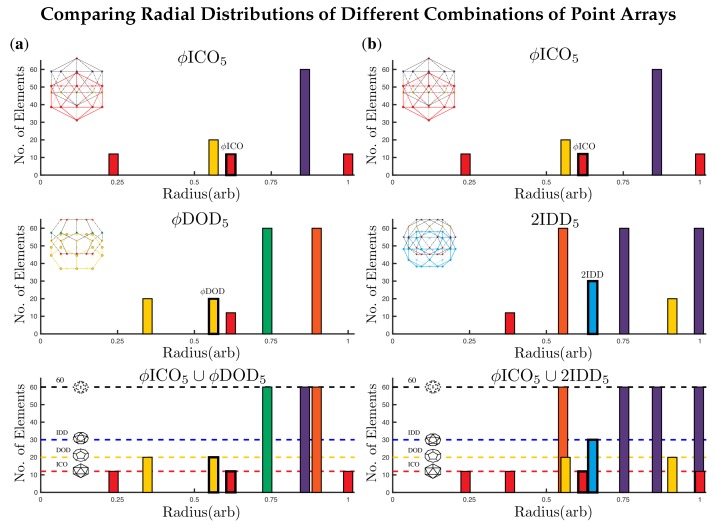
Here we examine two ways to form double point arrays with $\phi {\mathrm{ICO}}_{5}$. As before, the base polyhedra are outlined in bold. In (**a**) we examine the formation of $\phi {\mathrm{ICO}}_{5}\cup \phi {\mathrm{DOD}}_{5}$ which adds points above the icosahedral envelope of $\phi {\mathrm{ICO}}_{5}$ and could be used to locate a site for surface modification of the capsid. The cardinality of $\phi {\mathrm{ICO}}_{5}$ and $\phi {\mathrm{DOD}}_{5}$ is 116 and 172 respectively (Table 2); however each of these point arrays respectively generate the base of the other array, reducing the cardinality from 288 points to 256 points, with 8 radial levels, rather than 10. We also see the formation of (**b**) $\phi {\mathrm{ICO}}_{5}\cup 2{\mathrm{IDD}}_{5}$ which adds surface points near the same radial level as the original gauge points of $\phi {\mathrm{ICO}}_{5}$. The cardinality of $2{\mathrm{IDD}}_{5}$ is 242 and none of the points overlap $\phi {\mathrm{ICO}}_{5}$. Therefore, the total cardinality of the double point array is 358 and the number of radial levels is 11. Double base point arrays can be used in a variety of applications, e.g., they can provide a more complete radial description of a capsid, indicating geometric constraints on all proteins, or they could provide locations where ligands or decoration proteins could be added to meet these new conditions, or even suggest precisely where internal drugs should be placed to not disrupt the stability of the capsid.

**Figure 8 viruses-12-00467-f008:**
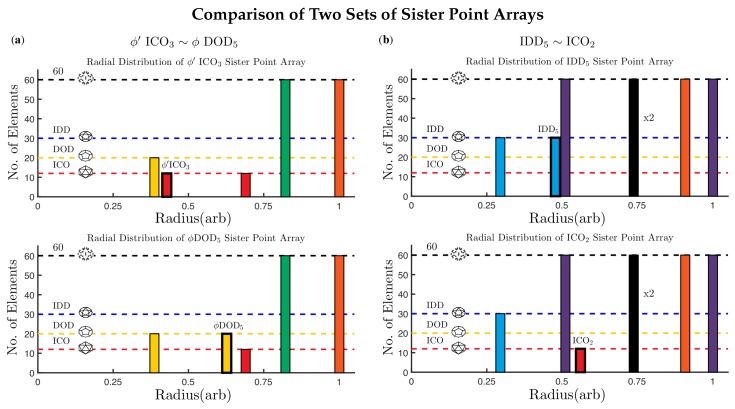
The radial distribution of point clouds for sister point arrays are identical, except for the base arrays (shown in bold outline). In (**a**) $\phi^{\prime }{\mathrm{ICO}}_{3}$ is identical to $\phi {\mathrm{DOD}}_{5}$ except for their two bases $\phi^{\prime }\mathrm{ICO}$ and ϕDOD. Please note that these point arrays have different affine extensions, so they cannot be combined. In (**b**) ${\mathrm{IDD}}_{5}$ is identical to ${\mathrm{ICO}}_{2}$ except for IDD and ICO. The bulk points at r∼0.75 have two sets of 60 points one on the left and one on the right side of the 5-2 great circle in the AU. While these points have the same radius, they are not equivalent locations.

**Figure 9 viruses-12-00467-f009:**
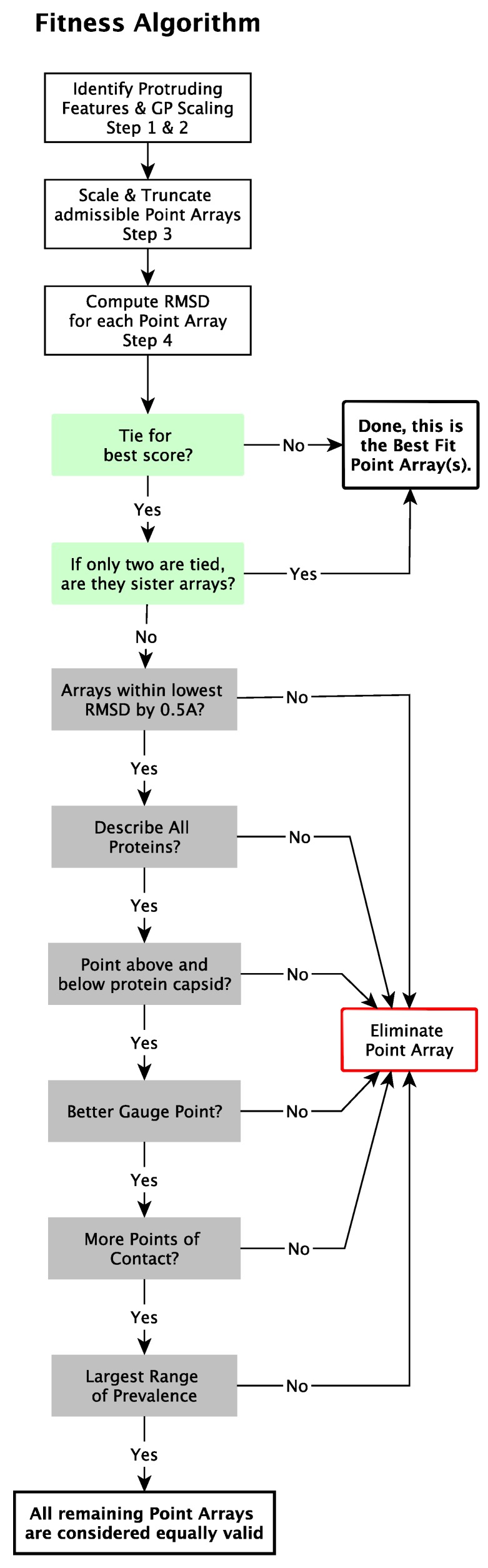
Schematic representation of the fitness algorithm for point array matching to viral capsids. Most viruses only have a few point arrays that have a low RMSD. Many ties are due to sister point arrays.

**Figure 10 viruses-12-00467-f010:**
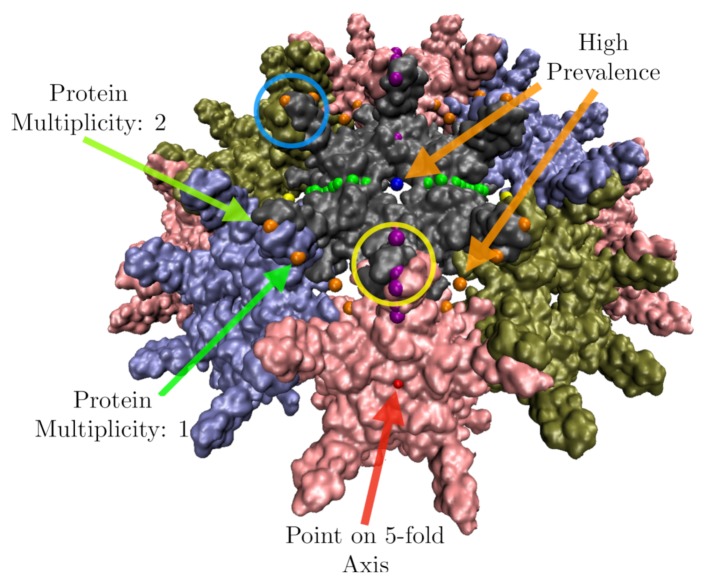
Features of the point array fitting algorithm. Here we have docked the gauge points on HBV, a *T* = 4 virus with dimeric protrusions. We only consider the point arrays with gauge points nearest the most radially distal protruding features, e.g., the nearest gauge points to the protruding dimers are GP: 17, 18 and 19 (yellow circles) and 4 and 5 (blue circles), see Table 2. In this example, there are 3 gauge points that would have fallen through the capsid surface when docked, which were instead stopped at their minimum distance to the surface. We see that most points sit on the surface of the proteins, and some on the symmetry axes in between proteins. Points which are near more than one protein are weighted more in the RMSD, through protein multiplicity $p_{i}$, see Equation (Equation 2). High multiplicity always occurs on symmetry axes e.g., Gauge Point 1 (red) on the 5-fold axis would count 5 times. There are several points which would lead to a high prevalence, i.e., they do not intersect proteins over a large range of radial scaling, including Gauge Point 15 (blue) which is on a 2-fold axis and Gauge Point 3 (orange). In this example, these points would lead to a poor RMSD fit, as they are each several angstroms away from the surface.

**Figure 11 viruses-12-00467-f011:**
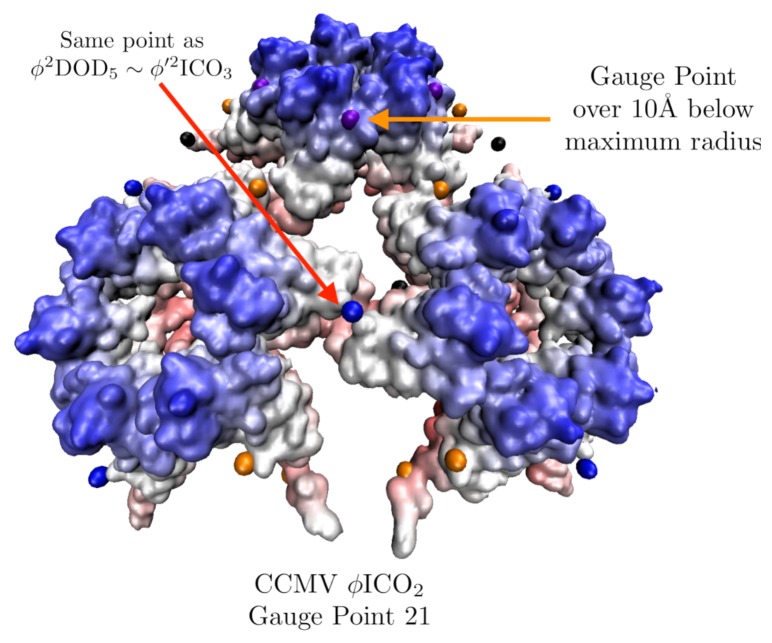
A section of the swollen CCMV capsid radially colored and fitted with $\phi {\mathrm{ICO}}_{2}$. In our fitness measure we consider all gauge points which are near protrusions. In this example, we see how a point array can have a low RMSD value, but not match the protruding features well and therefore be excluded later. In this case Gauge Point 21 is 10 Å below the most external atoms, and the other gauge points which sit atop the blue sections are the better fit.

**Figure 12 viruses-12-00467-f012:**
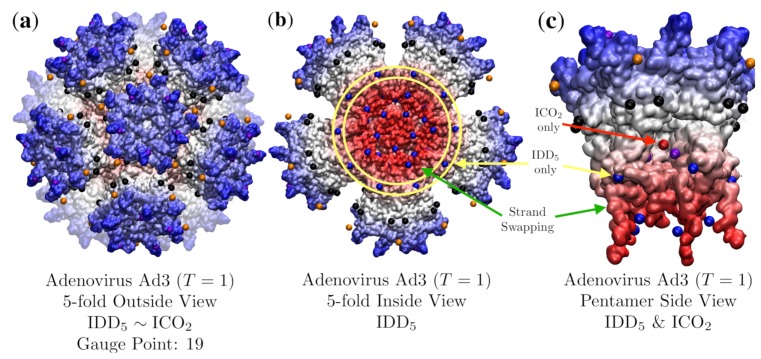
(**a**) Adenovirus Ad3 Dodecahedron is composed of a penton base (Ad3, *T* = 1, 4aqq [20]) and is best fitted by ${\mathrm{IDD}}_{5}$ with an RMSD of 3.7 Å and Gauge Point 19. Ad3 is not equally co-fit by the sister point array ${\mathrm{ICO}}_{2}$. The radial histogram of these sister point arrays appear in Figure 8b. It is noteworthy that the entire point array was used in the fitness algorithm, without any radial cutoff. While the gauge points and interior points agree well with the protein surface, there are several points (orange and black) that float a few angstroms off the surface. (**b**) The interior view of the capsid. Ad3 is stabilized by strand-swapping between neighboring pentamers, which occurs across the 2-fold axes between the IDD base (blue points encircled in yellow) of ${\mathrm{IDD}}_{5}$ and the innermost (lowest radial level) points (blue) which sit on the underside of the strand-swapping. (**c**) A side view of one of the pentamer subunits with a section of protein removed to show the interior 5-fold point (red) contained by ${\mathrm{ICO}}_{2}$ only and the interior 2-fold points (blue) contained by ${\mathrm{IDD}}_{5}$ only. These 2-fold points (blue) provided by ${\mathrm{IDD}}_{5}$ only, are situated at the major point of contact between the adjacent pentameric units, above the strand-swapping. As the two sister point arrays are identical except for these points (Figure 8), they are the deciding factors on which RMSD is lower. Here the ${\mathrm{ICO}}_{2}$ point (red) is sitting in an empty pocket, making the RMSD 4.9 Å 1.2 Å higher than ${\mathrm{IDD}}_{5}$ with an RMSD of 3.7 Å.

**Figure 13 viruses-12-00467-f013:**
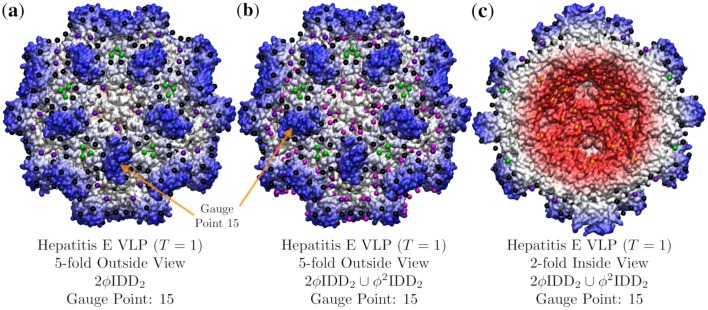
A demonstration of how point array fits may be improved using double base point arrays. In (**a**) we see *T* = 1 Hepatitis E virus-like particle (*T* = 1, 3hag [19]) fitted only by $2\phi {\mathrm{IDD}}_{2}$ with an RMSD of 3.2 Å and Gauge Point 15. This point array however does not have any points on the interior surface of the capsid. In (**b**) we combine $2\phi {\mathrm{IDD}}_{2}$ with the smaller radius point array $\phi^{2}{\mathrm{IDD}}_{2}$, shown here as magenta points, irrespective of their geometric locations deviating from Table 1. These new points sit perfectly on the surface and add interior surface locations and lower the RMSD to 2.8 Å. In (**c**) we see the interior of our capsid, $\phi^{2}{\mathrm{IDD}}_{2}$ shown in orange along the 5-2 great circle. The point array fitting of the capsid interior can also be seen in Figure 14.

**Figure 14 viruses-12-00467-f014:**
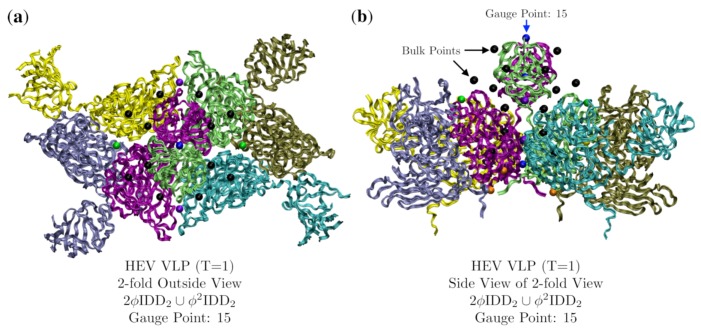
Exhibiting the point array description of Hepatitis E virus-like particle (*T* = 1, 3hag [19]) using a ribbon view with each protein colored separately. In (**a**) we are viewing down a 2-fold axis, the gauge point is 15 (blue) on the 2-fold axes. We can see the many point array elements nestled in and around the protein surface. In (**b**) we are looking at the same section from the side view, and you can see how the proteins fit neatly between the point arrays. The lowest radial points (orange) were added by $\phi^{2}{\mathrm{IDD}}_{2}$.

**Figure 15 viruses-12-00467-f015:**
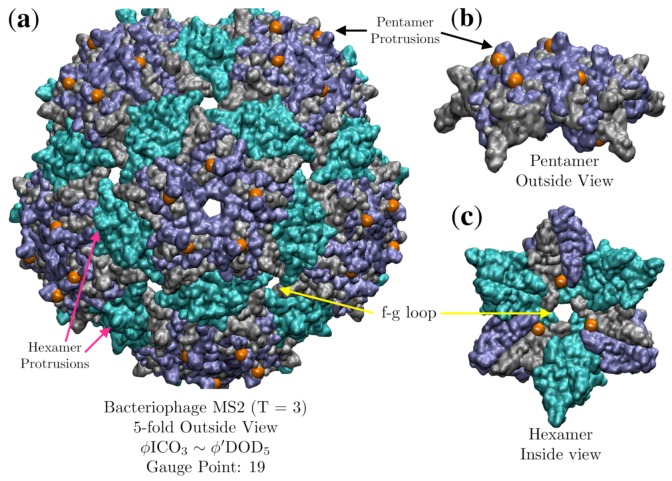
(**a**) Bacteriophage MS2 (*T* = 3, 2ms2 [29]), viewed down the 5-fold axes, is fitted best by $\phi {\mathrm{ICO}}_{3}∼\phi^{\prime }{\mathrm{DOD}}_{5}$ with an RMSD of 0.7 Å and Gauge Point 2. Equivalent capsid proteins are colored blue, gray and cyan. Though this fit uses only 2 of the 5 radial levels of the point arrays, its agreement with experiments is remarkable, as it is nearly impossible to substitute different amino acids near the gauge points (orange) [23,24]. (**b**) A perspective view of the gauge points sitting right next to the pentamer protrusions, which were found to be nearly immutable, suggesting this geometric location is critical to the stability of MS2. As MS2 is composed of 180 chemically identical proteins, the hexamer protrusions are similarly immutable, at least pre-assembly of the capsid. (**c**) It was shown that the f-g loop which is the inner surface of the hexamer hole was very mutable, which is also in agreement with the point array fit, as the restricted points are several angstroms away (orange). Directly above these points are the β-sheets.

**Figure 16 viruses-12-00467-f016:**
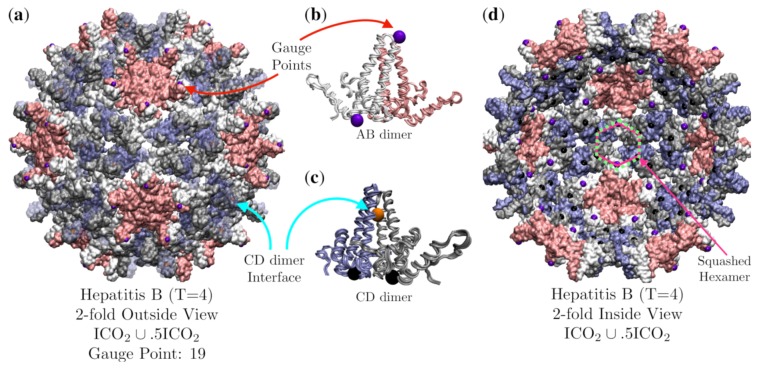
(**a**) A view down the 2-fold axis of Hepatitis B (*T* = 4, 1qgt [34]) which is composed of 120 homodimers. The AB dimers (A is pink and B is white) forms the 12 pentamers (pink) and the CD dimer (C is blue and D is gray) along with the B-protein form the 30 hexamer subunits (blue, gray and white). The capsid is best fitted by the double base point array point ${\mathrm{ICO}}_{2}\cup .5{\mathrm{ICO}}_{2}$ with an RMSD of 1.3 Å and Gauge Point 19 (purple) located on the AB pentamer dimers. There are also points (orange) located at the internal interface of the CD dimer (Semi-translucent blue proteins for clarity). (**b**) The AB dimer is bounded above (purple) and below (black) by point arrays elements. (**c**) The CD dimer of the hexamers has an orange point on the line of contact between the two α-helices of the CD dimer and rest atop two bulk points (black) on the bottom of each protein of the dimer. The CD dimer is more flexible than the AB dimer [39]. (**d**) Looking out from the interior of the capsid along a 2-fold axis, we see a slightly squashed hexamer (white, gray and blue) centered on 2-fold axis. Each hexamer is in contact with four interior bulk points (black) and two interior 5-2 GC points (purple). As such, the hexamers do not have local 3-fold symmetry, which is consistent with the point array constraints. For comparison, a squashed hexagon is shown as a solid line (pink) and an ideal hexagon is shown as a dotted line (green). Note that even though this point arrays was generated from two base icosahedra (ICO∪.5ICO), all of the 5-fold points have been truncated as they are more than 4 Å below the interior capsid.

**Figure 17 viruses-12-00467-f017:**
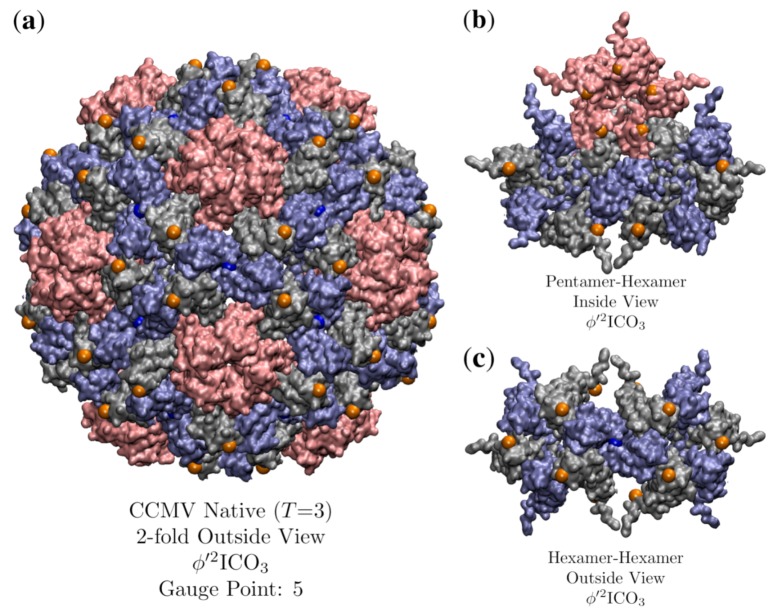
(**a**) A look down the 2-fold axes of CCMV (*T* = 3, 1cwp [30]) which is best fitted by $\phi^{\prime 2}{\mathrm{ICO}}_{3}$ with an RMSD of 0.7 Å and Gauge Point 5. The pentamers are shown in pink and the hexamers are gray and blue. (**b**) Inside the capsid view of the points as they coordinate the pentamer and hexamer meeting. (**c**) Outside view of the hexamers. The pentamer proteins only make contact with one point, though the hexmamer unit makes contact with 3 points, above, below and at their junction.

**Figure 18 viruses-12-00467-f018:**
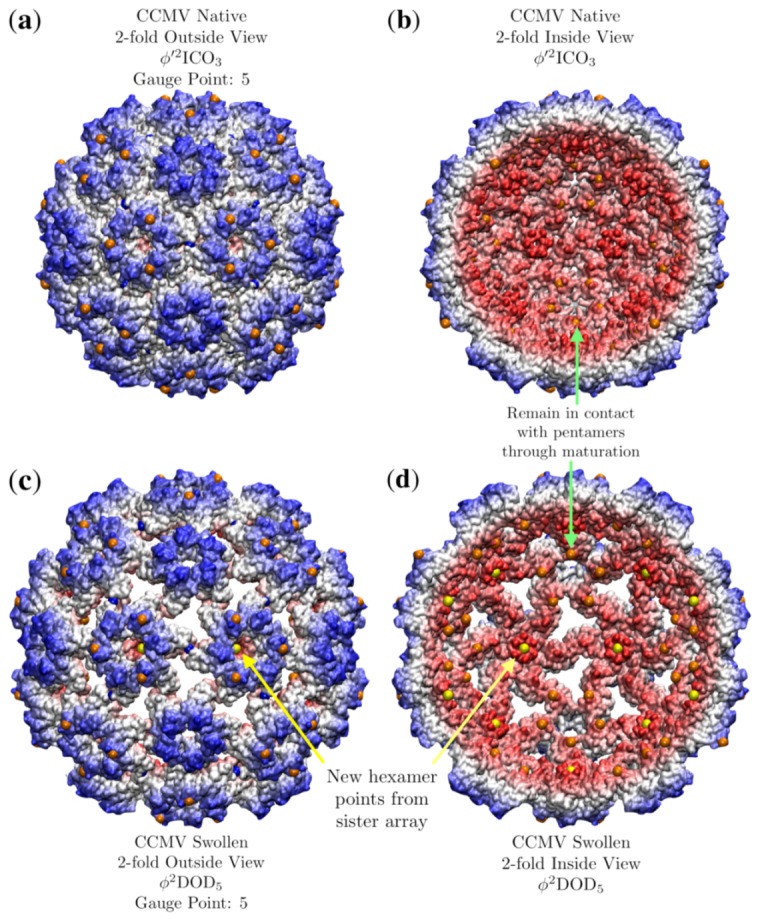
A comparison of the structural changes of CCMV (T=3) which occur during maturation using point arrays. Each capsid is radially colored. (**a**) Exterior view of CCMV native state, showing the prominent role that hexameric features play in determining the point array fit. (**b**) The interior of CCMV natives state. There are point array locations (orange) in contact with the interior surface of the pentamer, as was seen in Figure 17. (**c**) The swollen CCMV capsid is approximately a uniformly scaled native state, though the orientation of the hexamer chains rotate slightly and the center of the hexamer opens up at the top and closes at the bottom, allowing the sister point array $\phi^{2}{\mathrm{DOD}}_{5}$ to fit the capsid. The overall effect of this reorganization is to create holes in the protein capsid. (**d**) The interior view of the mature capsid showing how the hexamer unit changes to accommodate a new point (yellow) after maturation.

**Figure 19 viruses-12-00467-f019:**
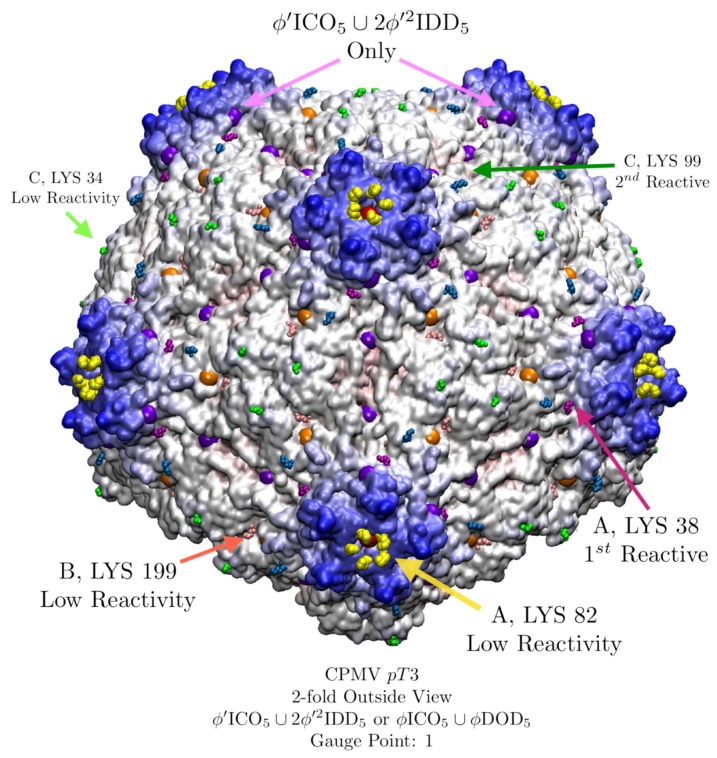
CPMV (pT3, 1ny7 [32]) has five solvent exposed exterior lysine residues found to be reactive which are labeled here by the protein domain they occupy [17]. There are two point arrays $\phi^{\prime }{\mathrm{ICO}}_{5}\cup 2\phi^{\prime 2}{\mathrm{IDD}}_{5}$ and $\phi {\mathrm{ICO}}_{5}\cup \phi {\mathrm{DOD}}_{5}$ that fit equally well with an RMSD of 1.5 Å and Gauge Point 1. This agreement is not surprising, as their truncated point arrays are identical except for a single point (purple) along the 5-2 GC. The geometric restrictions of the point arrays are in remarkable agreement with the reactivity data [17], explaining at least 4 of the 5 reactivity sites (Table 5).

**Figure 20 viruses-12-00467-f020:**
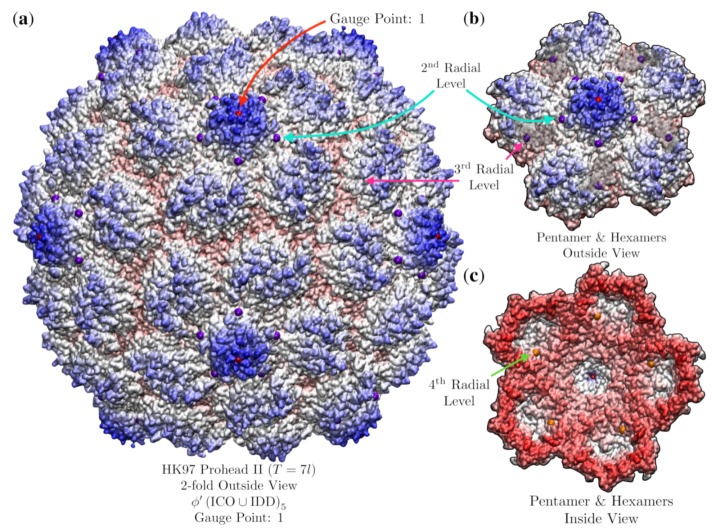
Point arrays may predict a loss of local symmetry of hexameric subunits for Bacteriophage HK97 Prohead II (T=7l, 3e8k [37]), as the hexamer subunits have different geometric restrictions at non-equivalent locations. (**a**) This capsid is best fitted by $\phi^{\prime }{\left(\mathrm{ICO}\cup \mathrm{IDD}\right)}_{5}$ with an RMSD of 1.8 Å. The pentamer protrudes at Gauge Point 1, as do many *T* = 7 viruses, and then the surface bows inward to fit beneath the 2nd radial level (purple). (**b**) There are small pockets within the hexamers, where the 3rd radial level along the 5-3 GC sits just below the surface of the hexamer (purple), the atoms above the pocket have been made translucent. (**c**) Inside view, each hexamer is bounded on the inside by the 4th radial level along the 5-3 GC point (orange). The pentamers and hexamers are each bounded by two different points within the AU.

**Table 1 viruses-12-00467-t001:** The color scheme used for point array elements throughout this paper. Each color specifies the type of geometric location, e.g., 5-fold points are red, 2-fold points are blue, and purple are the points on the great circle between the two (Figure 1).

**Location**	**Color**
5-fold	Red
5-3 GC	Orange
3-fold	Yellow
3-2 GC	Green
2-fold	Blue
2-5 GC	Purple
Bulk	Black
Origin	Teal

**Table 2 viruses-12-00467-t002:**
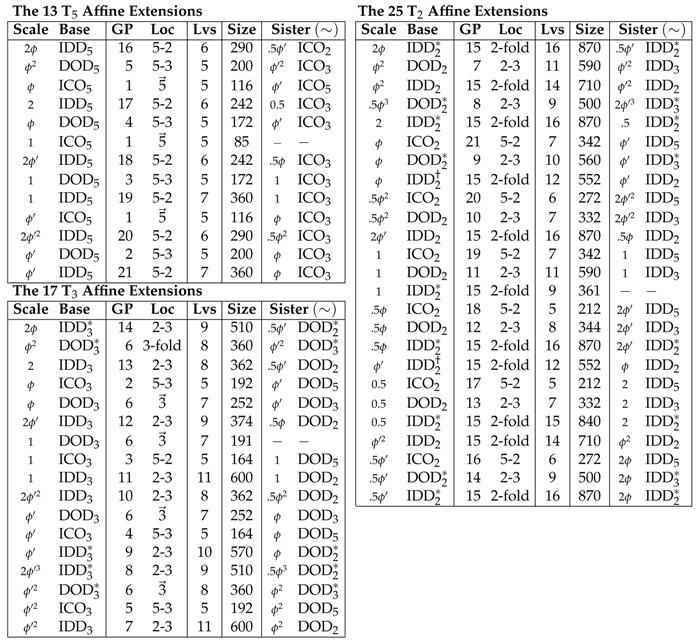
The 55 admissible point arrays [1,2], grouped by extension vectors ${\vec{T}}_{5},{\vec{T}}_{3}$ and ${\vec{T}}_{2}$, ordered from largest to smallest relative radii. The initial scaling of the base point arrays before affine extension is as shown. The gauge points and their location on the great circles are listed (Figure 1). The number of distinct radial levels, size of the full point array (cardinality) and sister point arrays are also given (3 arrays do not have sister arrays). While all sister arrays have identical point clouds, the two point arrays which are identical are indicated with †. The 14 point arrays with only a single radial level located on the icosahedral symmetry axes are indicated with an asterisk *. Three point arrays ${\mathrm{ICO}}_{5},{\mathrm{DOD}}_{3}$ and ${\mathrm{IDD}}_{2}$ have an element at the origin, which is degenerate. Note that $\phi^{\prime }=-1/\phi$; however all of the base polyhedra are invariant under multiplication by −1, therefore we only report the absolute value of the scaling.

**Table 3 viruses-12-00467-t003:** Here we present the results of our fitness algorithm of 16 viruses, considering RMSD scores, gauge points (GP), and points of contact (NAU). All fits are decided as in Figure 9. Most RMSD fits are at least better than 0.5 Å than the next best array. Several viruses are equally well fit by a point array and its sister array. When both sister point arrays are equally valid, e.g., ${\mathrm{ICO}}_{3}$ or ${\mathrm{DOD}}_{5}$ of STMV, both are listed. If two point arrays are tied for best, that is stated explicitly. There are two cryo-EM fit solutions for CCMV (swln1) in the swollen state, we analyzed ccmvswln1 [25]. All structures analyzed in this paper were obtained from the Viper Database [21].

Best Fit Point Arrays with RMSD Values						
**Name**	***T***	**PA**	**RMSD (Å)**	**GP**	**NAU**	**PDBID**
Adenovirus Ad3 Dodecahedron	1	${\mathrm{IDD}}_{5}$	3.7	19	7	4aqq [20]
Hepatitis E VLP	1	$2\phi {\mathrm{IDD}}_{2}\cup \phi^{2}{\mathrm{IDD}}_{2}$	2.8	15	26	3hag [19]
Infectious Bursal Virus	1	$.5\phi^{3}{\mathrm{DOD}}_{2}$	4.5	8	9	2gsy [18]
STMV	1	${\mathrm{ICO}}_{3}∼{\mathrm{DOD}}_{5}$	1.2	3	3	1a34 [26]
L-A Virus	2	$\phi {\mathrm{DOD}}_{5}\cup {\mathrm{DOD}}_{5}$	1.4	4	3	1m1c [27]
Bacteriophage GA	3	$\phi {\mathrm{ICO}}_{3}∼\phi^{\prime }{\mathrm{DOD}}_{5}$	0.2	2	2	1gav [28]
Bacteriophage MS2	3	$\phi {\mathrm{ICO}}_{3}∼\phi^{\prime }{\mathrm{DOD}}_{5}$	0.7	2	2	2ms2 [29]
CCMV Native	3	$\phi^{\prime 2}{\mathrm{ICO}}_{3}$	0.7	5	3	1cwp [30]
CCMV Swollen	3	$\phi^{2}{\mathrm{DOD}}_{5}$	2.7	5	4	swln1 [25]
Tobacco Necrosis Virus	3	$\phi^{\prime }{\mathrm{ICO}}_{5}\cup 2\phi^{\prime 2}{\mathrm{IDD}}_{5}$	0.9	1	5	1c8n [31]
Cowpea Mosaic Virus (CPMV)	pT3	$\phi^{\prime }{\mathrm{ICO}}_{5}\cup 2\phi^{\prime 2}{\mathrm{IDD}}_{5}$	1.5	1	5	1ny7 [32]
Helicoverpa (HASV)	4	${\mathrm{IDD}}_{5}\cup \phi^{\prime }{\mathrm{ICO}}_{5}$	1.1	19	6	3s6p [33]
Hepatitis B	4	${\mathrm{ICO}}_{2}\cup .5{\mathrm{ICO}}_{2}$	1.3	19	5	1qgt [34]
Nudaurelia Capensis ω Virus	4	${\mathrm{DOD}}_{5}\cup \phi^{\prime }{\mathrm{ICO}}_{5}$	1.5	3	5	1ohf [35]
Bacteriophage P22 Mature	7l	$\phi {\mathrm{ICO}}_{5}\cup \phi {\mathrm{DOD}}_{5}$	0.8	1	3	5uu5 [36]
HK97 Prohead II	7l	$\phi^{\prime }{\mathrm{ICO}}_{5}\cup \phi^{\prime }{\mathrm{IDD}}_{5}$	1.8	1	4	3e8k [37]

**Table 4 viruses-12-00467-t004:** At first glance, there appear to be several point arrays tied for the best fit of the swollen CCMV particle; however 3 of these were excluded based on poor agreement with the gauge points (Figure 11).

CCMV Swollen (T=3)				
PA	RMSD	GP	NAU	Notes
$\phi^{\prime 2}{\mathrm{ICO}}_{3}$	1.9	5	3	
$\phi {\mathrm{ICO}}_{2}\cup .5\phi^{2}{\mathrm{ICO}}_{2}$	2.2	21	6	excluded
$\phi {\mathrm{ICO}}_{2}\cup 2\phi^{\prime }{\mathrm{IDD}}_{2}$	2.3	21	7	excluded
$\phi {\mathrm{ICO}}_{2}\cup \phi {\mathrm{IDD}}_{2}$	2.6	21	8	excluded
$\phi^{2}{\mathrm{DOD}}_{5}$	2.7	5	4	

**Table 5 viruses-12-00467-t005:** A comparison of the reactivity of the 5 solvent exposed lysine residues of CPMV [17] and the geometric restrictions imposed by the point arrays. An X indicates that the point array is unlikely to permit changes at this location and a + indicates no specific geometric restriction. We also indicate our naive expectation of capsid reactivity based on good solvent accessibility (+) and lack of steric hindrance (–). Both point arrays indicate that A LYS 82 and B LYS 199 should have low reactivity as there is a point array element nearly atop each, implying that the site is a key structural feature, similar to the loop protrusions of MS2. Both point arrays also agree that C LYS 99 should have no restrictions on reactivity. While $\phi^{\prime }{\mathrm{ICO}}_{5}\cup 2\phi^{\prime 2}{\mathrm{IDD}}_{5}$ is the best fit point array, it indicates that the most reactive site A LYS 38 should not be reactive; however there is a loophole here as $\phi {\mathrm{ICO}}_{5}\cup \phi {\mathrm{DOD}}_{5}$ have the same truncated points except for this single geometric restriction, so CPMV can be reactive here as this location is not critical to the other point array stability. This reactive site A LYS 38 is a bit paradoxical as it is slightly hidden and less solvent exposed, yet it is very reactive. Lastly neither point array implies any restrictions for C LYS 34 though it has low reactivity and good solvent accessibility and low steric hinderance. It is possible that including the genomic material in the point array fitness would clarify this case [17].

CPMV (*pT* = 3) LYS Reactivity Comparison				
**Residue**	**Reactivity**	$\phi^{\prime }$ ${\mathrm{ICO}}_{5}\cup 2\phi^{\prime 2}$ ${\mathrm{IDD}}_{5}$ **or**	**Accessibility**	**Naive**
		ϕ ${\mathrm{ICO}}_{5}\cup \phi$ ${\mathrm{DOD}}_{5}$		**Prediction**
A LYS 82	Low	X/X	Solvent: +, Sterics: −	Good
A LYS 38	Highest	X/+	Solvent: −, Sterics: +	Poor
B LYS 199	Low	X/X	Solvent: +, Sterics: −	Good
C LYS 34	Low	+/+	Solvent: +, Sterics: −	Good
C LYS 99	Second	+/+	Solvent: −, Sterics: +	Poor

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
