# Peer review of "Unveiling the Hidden Rules of Spherical Viruses Using Point Arrays"

_viruses, 2020, doi:10.3390/v12040467_

Round 1

Reviewer 1 Report

This manuscript by David Wilson describes the development of point arrays to describe the geometry of virus capsids. By fitting and aligning capsids to point arrays, he finds that they can be used to explain, and perhaps predict, locations which will likely prohibit modification either via mutation or chemical reaction. Such an ability would be of great value to the community studying viruses and virus-like particles, and thus the work has the potential for substantial impact. However, the manuscript in current form makes it difficult to fully evaluate these potential applications, largely due to organizational issues:

The methods do not detail how the point arrays are generated in an easy-to-follow (or reproduce) manner. The description is instead spread across the methods and results sections, as well as figures and tables. For example, the results section refers to comparisons of a variety of virus capsids to the various arrays, and these are given in Table 3, but not described explicitly in the Methods. The RMSD fitting is also only described briefly in the text, and points to a figure for full details instead of providing the figure to capture the highlights of the method. Given the complexity of the topic, it is worth a more detailed text explanation of the operations carried out, not just for these two examples but for every operation used to arrive at the presented results. These should be organized by subheadings that describe each type of method. Results need to include the reasoning for undertaking a particular experiment/comparison/application. For example, Table 3 is first mentioned on Line 324 in a sentence focused on interpretation of the Table 3 findings, without first describing that several viruses were put through the process, how and why these were chosen, etc. This is one example but such an approach is needed throughout the results section in order to assist the reader (and me, as reviewer) in determining the validity of the finding, as well as how to extend such analysis to new systems. Lines 361-500 contain the majority of the intriguing results as alluded to in the introduction, abstract, and conclusion, but are missing in results section of the pdf. None of this section appears to be described in the Methods either. There are numerous typos/grammatical errors that should be corrected, as well as several abbreviations used without defining them first either in the text or legends of figures (eg AU, GC). It is difficult to represent 3-D analyses in 2-D format, so perhaps a supplemental section could include a step by step graphical description of how the algorithms are applied to one of the model viruses.

Reviewer 2 Report

This manuscript describes a set of geometric rules and constraints that can be used to explain and predict virus capsid structures and the location of features of importance to assembly and capsid integrity, taking as its starting point the work by Keef and Twarock (2009) and a previous publication (Wilson 2016). The author has obtained interesting results from his analysis of models systems such as MS2, Hepatitis E virus, and CCMV. As such, this work should be of interest to researchers who study virus assembly.

Unfortunately, the paper is extremely difficult to read and comprehend, at least to a non-mathematician like myself: Terminology and notation are introduced with little or no explanation that might make sense to a specialist, but that I fear will be out of grasp of most virologists, presumably the target audience for this paper. Moreover, the paper is poorly organized and seems rather sloppily put together. Several figures are out of order; the text is wordy and repetitive, with sections that mix theory, results and discussion; and the manuscript is full of typos and other mistakes.

In light of these concerns, I do not think this paper is a good fit for Viruses, and I cannot recommend it for publication in its present form. Nevertheless, I do think a study such as this could be of value to the virus structure community, and that a considerably rewritten manuscript could be considered in the future.

Specific comments follow:

line 21: “as locations that viruses should only be modified with care” is syntactically incorrect.

line 22: as well -> as well as

line 26-27: this statement is true only with modifications: Many viruses have 120 copies of a capsid protein, papilloma- and polyomaviruses have 360 (“T=6”) arranged as all pentamers, and others (adeno-, irido= and mimiviruses) have trimers in their hexavalent positions (in the case of adeno, 240 trimers + 12 pentamers = 780 protein copies placed on a pseudo-T=25 lattice).

line 38: mores -> more

line 39: sites -> site

line 41: Figures 7, 12, 11 and 14 are introduced out of order (7 1 2 3 4 5 6 12 8 9 11 10 14 12 13 15 16 17 18 19).

line 41 and other places: instead of using “see Figure X” use “(Figure X)”

line 45: “of them is”

line 49: “by [13]” should be referred to as “by Rochal et al. [13]”

line 55: use “Refs 3, 15”

line 59: what is “pT3”? “pseudo-T=3” presumably?

line 67: largest -> highest

line 69: “without almost nothing”

line 76-78: confusing sentence. Re-write.

line 83: notation [0,1,phi] etc. is not introduced.

Figure 1 legend: Matradenovirus à Mastadenovirus

line 85: syntax

line 86: polyherda -> polyhedral

line 91: unclear: “on a sphere modulo spatial inversions”

Formula 1: Unclear. What are a and b?

Figure 3: “a histogram the radii” Insert “of”

Figure 4 legend: point array point array

line 140: point -> points

line 172: arrays -> array, does à do

line 277-278: “maturation, which can … trigger maturation” doesn’t make sense.

line 281: constrains -> constraints; previous -> previously; extra “of”

line 266: hinderance -> hindrance

line 307: Discussion. This section should be renamed “Results” or “Results and Discussion” and merged with the previous section.

line 310: modoficatiosn

Table 3: bacteriophage -> bacteriophage

line 335: “we do not believe that it is not imperative” is unclear; capsid -> capsids.

line 247: utilize -> utilized

line 358: the past participle of fit is “fitted”

line 369: different -> difference

line 370: fit -> fitted

line 377-379: garbled sentence

line 381: “an excess of capsid protein is produced.” Presumably it is an excess of penton protein that is produced?

line 414: fit -> fitted

line 423: fit -> fitted

line 424: slightly differ -> differ slightly

line 439-440: garbled sentence

Reviewer 3 Report

The author presents an algorithm to map geometric constraints from affine extended symmetry groups on virus structure, and discusses the implications for a wide range of viral examples. The potential of these mathematical structures for understanding the geometric principles of virus architecture had not been fully exploited in the past, and this paper presents a valuable effort to extend the range of this modelling. In particular, it provides a refined algorithm for the mapping of these point arrays on viral structures that makes a valuable contribution to the field. It also draws some interesting conclusions from a biological point of view, e.g. it provides an explanation for outcomes of protein engineering in MS2 (l.396/398).

However, there are a number of points the author should address before publication:

  • Caption of Figure 5: “These double arrays can be interpreted as… “ – this needs more explanation.
  • Does the order in which the rules for tie breaking (l.233-241) are used impact on the outcome? Please comment on whether this has been checked in your examples, or give a reason why this is not required.
  • Please expand on your reasoning that “these locations may be connected to equilibrium locations for vibrational modes (l. 269)
  • 278/9: meaning of “new pseudo-sterically hindered geometric constraints” is not clear.
  • 283: “serve as key structural locations” – for what?
  • Claims around l.297-300 should be illustrated based on a concrete biological example in order to be understandable for a wider readership. Fig. 3, 5 and 6 are not sufficient.
  • 311: “successful” in what? Please be more precise.
  • 316: what does “larger points” mean?
  • 327: “leading us to predict that” – this should be made more precise: based on how many data sets etc…
  • 347: “appears to have utilised” – requires illustration by a picture
  • 352: “There bases” – which?
  • Around L.442: It should be mentioned that asymmetric distortions from symmetric constraint sets are possible, because the point arrays do not fully constrain capsid architecture.
  • 445/6: “The radial distribution… icosahedral symmetry.” – delete or rephrase

Minor points:

 -l.31: delete “was”

-l.46: is -> be

-l.69: delete “without almost nothing in common”

-l.74: limits -> constraints

-l.81: delete “which may offer additional stability to the viral capsid.” – please note that point arrays cannot do this…

-l.86: polyherda -> polyhedral

-l.111: table 2 -> Table 2

-l.114: arrays -> array

-Table 1 caption: use -> used

-l.157: the if -> that if

-l.170/1: have a sister array

-l. 267: meeting locations -> contact points

-l.271: please add “in the sense of Janner’s encasing forms” after “boundary constraints”

-l.272: replace one of the occurrences of “sense”

-l.272: “can’t” should be replaced by “cannot”

-l.273/l.275: “location to meet” and “will meet” should better be described as “contact” or “contacting”

-l.281: previously known; constraints; of of

-l.305: off -> of

-l.326: we have

-l.335: double negation… delete both “not”s

-l.356: replace “location if the capsid did not use it” by “constraint from the overall constraint set”

-l.358: except for

-l.359: difference in

-l.363: perfect, as…

-l.378: you -> one

-l.386: adhering to. In…

-l.404: should still be possible

-l.417: lying down on -> parallel to

-l.420: it should

-l.424: spaces!

-l.428: boundary conditions of -> geometric constraints on

-l.436: axes

-l.439: has been suggested

-l.462: native state; what does “confirmations” mean here?

-l.472: delete “now”

-l.482: does not

-l.485: full stop missing

-l.502/3: Even if it were…

-l.546: may may

Round 2

Reviewer 2 Report

The revised version of this manuscript by Dr. Wilson has been greatly improved over the original. The new organization of the paper has made a huge difference on readability. While the theoretical derivation of the architectural principles of viruses is still very complex, most structural virologists will still find the results on various systems interesting and informative, and those who want to delve into the mathematical theory can now find this in the clearly separated methods section. The background on icosahedral symmetry and point arrays has been much better explained in the revised manuscript and should be accessible to those who take the time to read it carefully. With the many changes and improvements, I find that the manuscript is now acceptable for publication in Viruses. I have only a few minor changes/corrections:

In Figures 1 and 2 the terms IDD, DOD and so on precede their eventual definition in Figure 3. This could probably be solved simply by referring to Figure 3 in the legends to Figures 1 and 2.

The text in several figures is too small, especially Figure 1c (axis labels and gauge point markers), Figure 4b, Figure 6a and b, and Figure 7.

Line 340: points arrays -> point arrays.

Line 342: Add period after “circles” and start a new sentence with “For examples…”

Line 245: Remove “On one hand” as there appears to be no “on the other hand”

Line 400, 418, 482 and possibly other places throughout the manuscript: The passive present tense of the transitive verb “to fit” is “is fitted”, not “is fit”.

Line 423-424: Move “however” to after “perfect”

Line 522: is -> are. “Crystallization data” should be “crystal structure” or “crystallographic data” or something like that.

Line 593: “nor” should be “or” or “nor by”
